# MECHANISTIC DETECTION AND MITIGATION OF HALLUCINATION IN LARGE REASONING MODELS

**Zhongxiang Sun[1][2], Qipeng Wang[1], Haoyu Wang[1], Xiao Zhang [1], Jun Xu[1]***
[1]Gaoling School of Artificial Intelligence, Renmin University of China, Beijing, China
[2]Institute for the Future of Humanity, Renmin University of China – Westlake University, China

sunzhongxiang@ruc.edu.cn, wqp@ruc.edu.cn, wanghaoyu0924@ruc.edu.cn,
zhangx89@ruc.edu.cn, junxu@ruc.edu.cn,

## ABSTRACT

Large Reasoning Models (LRMs) have shown impressive capabilities in multi-step reasoning tasks. However, alongside these successes, a more deceptive form of model error has emerged—*Reasoning Hallucination*—where logically coherent but factually incorrect reasoning traces lead to persuasive yet faulty conclusions. Unlike traditional hallucinations, these errors are embedded within structured reasoning, making them more difficult to detect and potentially more harmful. In this work, we investigate reasoning hallucinations from a mechanistic perspective. We propose the **Reasoning Score**, which quantifies the depth of reasoning by measuring the divergence between logits obtained from projecting late layers of LRMs to the vocabulary space, effectively distinguishing shallow pattern-matching from genuine deep reasoning. Using this score, we conduct an in-depth analysis on the ReTruthQA dataset and identify two key reasoning hallucination patterns: early-stage fluctuation in reasoning depth and incorrect backtracking to flawed prior steps. These insights motivate our **R**easoning **H**allucination **D**etection (**RHD**) framework, which achieves state-of-the-art performance across multiple domains. To mitigate reasoning hallucinations, we further introduce **GRPO-R**, an enhanced reinforcement learning algorithm that incorporates step-level deep reasoning rewards via potential-based shaping. Our theoretical analysis establishes stronger generalization guarantees, and experiments demonstrate improved reasoning quality and reduced hallucination rates. The source code and dataset are available at:
https://github.com/Jeryi-Sun/Reasoning_Hallucination.

## 1 INTRODUCTION

Hallucination has long been a critical safety challenge for Large Language Models (LLMs). In this context, hallucination refers to outputs that appear fluent and coherent but are semantically inaccurate or lack factual grounding. With the advent of Large Reasoning Models (LRMs)—such as DeepSeek-R1 (DeepSeek-AI, 2025) and OpenAI's O-series (OpenAI, 2025)—AI systems have demonstrated unprecedented potential in solving complex real-world tasks. These models are typically trained with outcome-based reinforcement learning (RL) and explicitly generate multi-step reasoning traces prior to final answers.

Recent studies have uncovered a subtler form of hallucination emerging in LRMs (Lu et al., 2025; Vectara Research, 2025; OpenAI, 2025), which is referred to as **Reasoning Hallucination**. Unlike traditional hallucinations, reasoning hallucinations are often embedded within logically coherent reasoning traces, making incorrect information more persuasive and harder to detect. This form of "plausible but incorrect" reasoning can elicit user trust, resembling the conjunction fallacy, where detailed yet misleading explanations are perceived as more credible than simpler ones (Tentori et al., 2004; Valmeekam et al.). Prior studies mainly assess the correctness of reasoning paths in standard Chain-of-Thought (CoT) tasks over relatively simple problems (Xu et al., 2024; Prasad et al., 2023; Li et al., 2024b), with limited investigation into the mechanisms of hallucinations in LRMs. Recent

---

*Corresponding author.

work has extended evaluation to long CoT generated by LRMs (He et al., 2025; Lu et al., 2025), yet remains focused on error identification rather than uncovering underlying causes from mechanistic perspective. However, directly analyzing model-generated traces can be misleading due to the subtle nature of reasoning hallucinations. The emergence of Latent CoT, where reasoning is embedded in hidden states rather than surface text, further obscures detection (Hao et al., 2024). These challenges call for probing the internal mechanisms behind reasoning hallucinations, enabling interpretable and robust hallucination detection.

Recent studies on the reasoning capabilities of LRMs (Mirzadeh et al., 2024; Yan et al., 2025) have shown that models often produce incorrect answers when their reasoning process relies on shallow pattern-matching rather than genuine deep reasoning. This mirrors findings in cognitive science, where human thinking patterns are closely linked to the emergence of cognitive illusions (Kahneman, 2011; Weis & Kunde, 2024). Inspired by these observations, we investigate reasoning hallucinations in LRMs through the lens of internal thinking patterns, where a central challenge is how to quantify whether a model is performing deep reasoning or merely matching surface-level patterns from training data. Prior mechanistic interpretability studies highlight a functional division within language models: early layers primarily transmit information, while later layers perform more complex reasoning over aggregated context (Nikankin et al., 2025; Chen et al., 2025a). Based on this insight, we introduce **Reasoning Score**, which measures the divergence between logits obtained from projecting late layers of LRMs to the vocabulary space. Through synthetic experiments, we validate the effectiveness of the Reasoning Score in measuring the depth of reasoning in LRMs, which reflects whether the model engages in shallow pattern-matching or deep reasoning (§ 3.1).

Building on the proposed reasoning score, we conduct extensive analyses on reasoning hallucinations using the ReTruthQA dataset. We identify three key patterns of reasoning hallucination: **Pattern #1**: large fluctuations in reasoning depth during the early steps, and **Pattern #2**: incorrect backtracking from later steps to earlier incorrect steps. We attribute these patterns to the presence of shallow pattern-matching and overthinking steps, which undermine the LRM's inherent abilities in self-verification and backtracking, ultimately leading to reasoning hallucinations (§ 3.2). Moreover, we observe that **Pattern #3:** overthinking steps exhibit a positive correlation between reasoning scores and perplexity, indicating spurious verification behaviors (§ 3.3). Based on these findings, we design the **R**easoning **H**allucination **D**etection (**RHD**) method, which significantly outperforms baselines across diverse domains in the reasoning hallucination detection dataset (§ 4.1).

We further investigate the underlying cause of shallow pattern-matching and overthinking steps in LRMs and attribute it to the outcome-based RL paradigm commonly used during training. This paradigm incentivizes correct final answers but neglects whether intermediate reasoning steps reflect deep and meaningful thinking. To address this challenge, we introduce a step-level deep reasoning reward based on the reasoning score and propose **GRPO-R**, a variant of Group Relative Policy Optimization (GRPO) (Shao et al., 2024; DeepSeek-AI, 2025) that incorporates potential-based reward shaping. GRPO-R encourages deep—but not excessive—reasoning during RL fine-tuning. Our theoretical analysis shows that GRPO-R leads to better generalization in outcome-based RL, and empirical results confirm that it improves reasoning accuracy compared to standard GRPO (§ 4.2).

## 2 RELATED WORKS

**Hallucination of Language Models.** Hallucination remains a fundamental safety concern for LLMs, and outcome-supervised LRMs (DeepSeek-AI, 2025; OpenAI, 2025) exacerbate this issue by generating logically flawed but persuasive reasoning traces, a consequence of reward-seeking behavior induced by outcome-based RL without step-level supervision (Chen et al., 2025b; Valmeekam et al.; Sun et al., 2025c). Detection approaches span uncertainty estimation (Kadavath et al., 2022; Ren et al., 2022), internal signal probing (Chen et al., 2024; Li et al., 2025b; 2024a), process-level critique models (He et al., 2025), and Process Reward Models (PRMs) (Zhang et al., 2025; Sun et al., 2025b), though challenges remain due to the deceptive nature of hallucinated traces and the poor generalization of PRM signals (Zheng et al., 2024b). We address this by conducting a mechanistic analysis of reasoning hallucinations and proposing a detection method grounded in internal model behavior.

**Mechanistic Interpretability.** Mechanistic interpretability (Ferrando et al., 2024; Elhage et al., 2021) explains model behavior by attributing predictions to internal components, e.g., attention heads

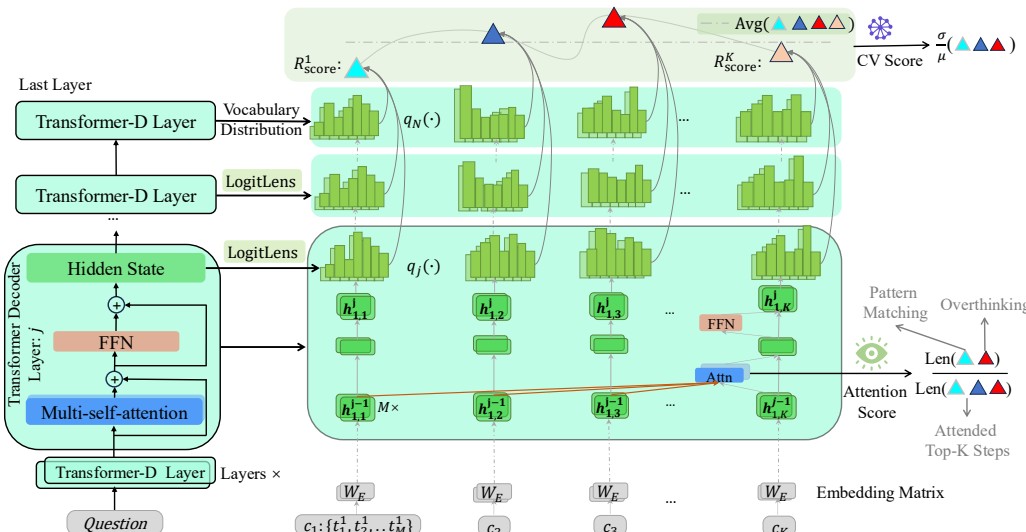

**Figure 1:** The illustration of the calculation processes for the Reasoning Score (Eq. 2), CV Score (Eq. 3), and Attention Score (Eq. 4).

contextualize token representations (Ferrando & Voita, 2024; Wu et al., 2024; Sun et al., 2026), while FFNs serve as knowledge storage (Geva et al., 2021). Intervention-based studies further reveal a division of labor across layers, where early layers transmit contextual information and later layers conduct complex reasoning (Chen et al., 2025a; Nikankin et al., 2025; Li et al., 2024c). These insights motivate our *Reasoning Score*, which quantifies hidden state shifts in later layers to capture thinking patterns and analyze reasoning hallucinations in LRMs.

## 3 EMPIRICAL STUDY OF REASONING HALLUCINATION

Our empirical study investigates the relationship between reasoning hallucinations and the thinking patterns of LRMs, where thinking patterns are quantified using a reasoning score derived from mechanistic interpretability. This analysis reveals key reasoning hallucination patterns and guides the design of more effective detection and mitigation strategies.

### 3.1 REASONING SCORE: MEASURING REASONING DEPTH IN LARGE REASONING MODEL

To determine whether a reasoning step is generated via shallow pattern matching or genuine deep reasoning, we propose a *Reasoning Score* inspired by mechanistic interpretability. Prior studies analyzing the internal mechanisms of language models reveal a layered functional division: early layers primarily transmit information, while later layers perform more complex reasoning over aggregated context to produce correct outputs (Stolfo et al., 2023; Nikankin et al., 2025; Li et al., 2024c; Sun et al., 2025a). Building on this insight, we define the reasoning score under the hypothesis that deeper reasoning is reflected by meaningful transformations in later-layer representations during generation.

Formally, a LRM-generated reasoning trace $C = [c_1, c_2, \ldots, c_K]$ consists of multiple reasoning steps, each associated with a step-level reasoning score $R_{\text{score}}^k$ that quantifies the depth of reasoning in step $c_k$. Each reasoning step $c_k = \langle t_1^k, \ldots, t_M^k \rangle$ is composed of $M$ tokens. The overall reasoning trace score $\mathcal{R}_{\text{score}}$ is represented as a sequence $[R_{\text{score}}^1, R_{\text{score}}^2, \ldots, R_{\text{score}}^K]$, capturing the model's reasoning dynamics across steps. As shown in Figure 1, each score is defined as the mean Jensen–Shannon divergence (JSD) between vocabulary distributions induced by hidden states from selected later layers and the anchor distribution from the final layer. To obtain the output distribution from each token hidden state $h_{m,k}^{(j)}$ of token $t_m^k$ at layer $j$, we apply the LogitLens (nostalgebraist, 2020), which projects each layer-normalized hidden state into vocabulary space via the unembedding matrix $\boldsymbol{W}_U$: $\text{LogitLens}(h_{m,k}^{(j)}) = \text{LayerNorm}(h_{m,k}^{(j)})\boldsymbol{W}_U$. This provides a layer-wise interpretation of token

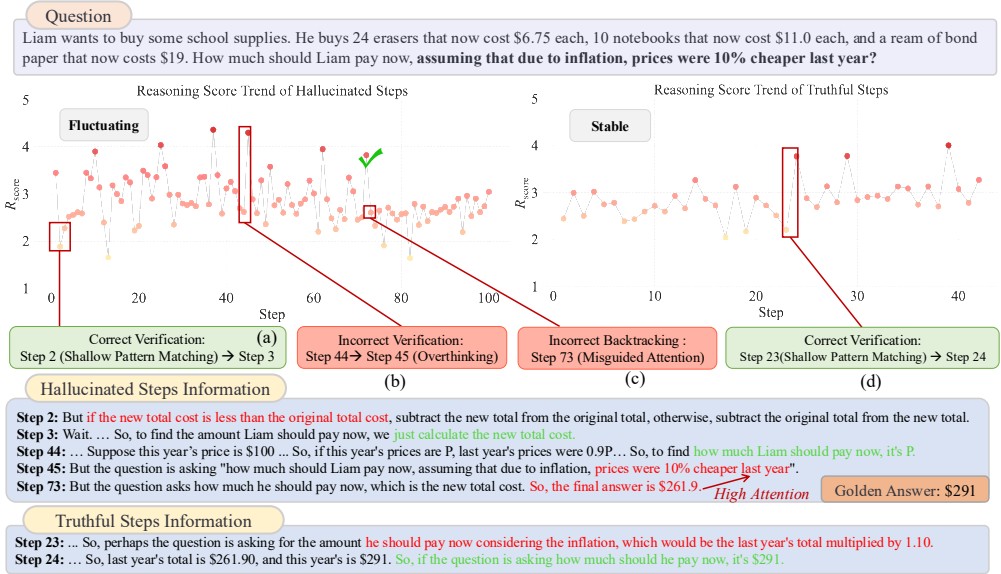

**Figure 2:** Case study from GSM-NoOp dataset Mirzadeh et al. (2024) on R1-7B. We sample both a hallucinated reasoning trace (left) and a truthful reasoning trace (right) for the same question as a preliminary analysis of reasoning hallucinations. Reasoning scores are scaled by $1e5$.

prediction behavior and has been widely adopted for interpreting LLM internal representations (Hanna et al., 2024; Zhou et al., 2024; Yu et al., 2023).

The final step-level Reasoning Score $R_{\text{score}}^k$ is computed as:

$$R_{\text{score}}^k = \frac{1}{|c_k|} \sum_{t_{m+1}^k \in c_k} \frac{1}{|\mathcal{J}|} \sum_{j \in \mathcal{J}} \text{JSD}\left(q_N\left(t_{m+1}^k\right), q_j\left(t_{m+1}^k\right)\right), \tag{1}$$

$$q_j\left(t_{m+1}^k\right) = \text{softmax}\left(\text{LogitLens}\left(h_{m,k}^{(j)}\right)\right), \quad j \in \mathcal{J}, \tag{2}$$

where $\mathcal{J}$ denotes the set of selected later layers and $q_N$ is the anchor distribution from the final layer.

Intuitively, a larger score $R_{\text{score}}$ indicates substantial transformation in output distributions within late layers, suggesting the model is actively engaging in deep reasoning by integrating earlier contextual information. In contrast, a smaller score implies distributional stability in late layers, indicating shallow pattern matching or heuristic-based processing without further reasoning, consistent with prior findings on the differential roles of early versus later layers.

**Validating the Reasoning Score with GSM-NoOp.** We validate whether the Reasoning Score faithfully reflects reasoning depth using GSM-NoOp (Mirzadeh et al., 2024), a GSM8K-derived dataset where semantically irrelevant but plausible No-Op phrases are injected into problems. Although these phrases do not alter the correct reasoning path, prior work shows that LRMs are often misled by them, revealing their reliance on shallow pattern matching (Mirzadeh et al., 2024). This makes GSM-NoOp a suitable testbed: if the Reasoning Score captures reasoning depth, then steps misled by No-Op phrases should yield lower scores. We validate this using correct outputs from DeepSeek-R1-Distill-Qwen-7B (R1-7B) to avoid confounds from hallucinated traces. Misled steps are labeled via GPT-4o. As GSM-NoOp is not publicly available, we re-implement a compatible version following the original paper's methodology, with prompts and details provided in Appendix E.

**Results.** Our empirical results in Figure 3 (a) show that reasoning steps misled by No-Op phrases consistently receive significantly lower Reasoning Scores compared to non-misled steps. This supports our hypothesis that the Reasoning Score effectively captures shallow pattern-matching behavior and serves as an indicator of whether a model is engaging in deep reasoning.

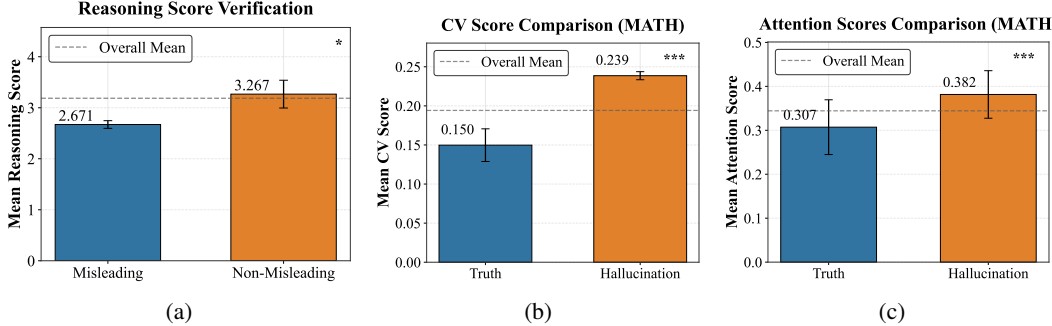

**Figure 3:** (a) Reasoning Score validation on GSM-NoOp. (b) Evaluation of Pattern #1 (early fluctuations), and (c) Pattern #2 (misguidedly attention) on ReTruthQA. Asterisks indicate statistical significance based on a t-test: * for $p$-value < 0.05, and *** for $p$-value < 0.001.

## 3.2 REASONING HALLUCINATION ANALYSIS BASED ON REASONING SCORE

In this section, we leverage the mechanistically derived Reasoning Score as a proxy for the thinking patterns of LRMs and investigate its relationship with the emergence of reasoning hallucinations. We begin with a preliminary analysis to identify characteristic patterns associated with hallucinated reasoning traces. We then analyze the generality of these patterns across domains using the ReTruthQA dataset, and further examine the underlying mechanism that leads LRMs to exhibit such behaviors.

### 3.2.1 CASE ANALYSIS ON GSM-NOOPS

In this section, we conduct a preliminary analysis using the LRM R1-7B on a question from GSM-NoOp (Mirzadeh et al., 2024), where a "NoOp" statement is appended to the end of a math problem. To enable controlled comparison of reasoning hallucination patterns, we sample both a truthful and a hallucinated response from R1-7B on the same question. Figure 2 presents the question along with step-level reasoning scores $R_{\text{score}}$, which quantify the depth of thinking at each step.

We observe that when the model generates reasoning steps that attend to the added NoOp content, these steps typically receive lower $R_{\text{score}}$, which in turn triggers the model's *Self-Verification* mechanism (Li et al., 2025a), producing later steps with higher $R_{\text{score}}$ that attempt to correct the earlier deviation (e.g., (a) and (d) in Figure 2). However, in the hallucinated reasoning trace, we also observe *overthinking* phenomena—steps with excessively high $R_{\text{score}}$ that incorrectly revise the previous correct reasoning steps (e.g., (b) in Figure 2). These hallucinated traces contain more shallow pattern-matching and overthinking steps, resulting in an overall unstable reasoning trajectory. From this case study, we identify the reasoning hallucination **Pattern #1**: hallucinated traces typically exhibit large fluctuations in reasoning score, especially during the early steps of the process.

Furthermore, we observe that even when the model briefly arrives at correct intermediate steps, it often fails to maintain this correctness. In later steps, it performs *Incorrect Backtracking*, attending to earlier shallow or overthinking steps, ultimately leading to hallucination (e.g., (c) in Figure 2). This motivates the reasoning hallucination **Pattern #2**: in the later stages of reasoning, the model tends to misguidedly attend to earlier hallucinated steps, either shallow or overthinking, making it difficult to correct earlier errors and leading to hallucinated reasoning.

### 3.2.2 REASONING HALLUCINATION PATTERN ANALYSIS

In this section, we validate the two reasoning hallucination patterns identified in preliminary analysis(§ 3.2.1): **Pattern #1**: large fluctuations in reasoning scores during early steps, and **Pattern #2**: incorrect backtracking to earlier hallucinated reasoning steps in later stages. We aim to assess whether these patterns generalize across broader domains and tasks. To this end, we conduct experiments on the ReTruthQA dataset using the R1-7B model. ReTruthQA covers three reasoning domains: `Math`, `Science`, and `MultiHopQA` (Details in § 5.1). For each domain, we construct two balanced subsets using gold hallucination labels: one with hallucinated traces and one with truthful traces.

To evaluate **Pattern #1**, we measure the fluctuation of reasoning depth in the early phase of reasoning using the Coefficient of Variation (**CV Score**) (Everitt, 1998), a standard metric for quantifying

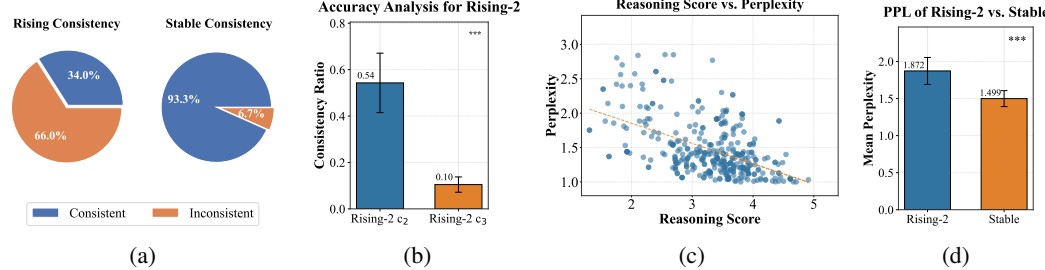

**Figure 4:** Analysis of Pattern #1: (a) Consistency Analysis (**Q1**); (b) Accuracy Comparison in Rising-2 triples (**Q2**); (c) Reasoning score vs. perplexity and (d) Perplexity of Rising-2 vs. Stable (**Q3**).

sequence variability (shown in Figure 1). Specifically, we focus on the first $\lceil K/r \rceil$ steps of the reasoning trace $\mathcal{C} = \langle c_1, c_2, \dots, c_K \rangle$, and define: $\mathcal{R}_{\text{score}}^{\text{early}} = \left[ R_{\text{score}}^1, R_{\text{score}}^2, \dots, R_{\text{score}}^{\lceil K/r \rceil} \right]$, where $r > 1$ is a constant controlling the size of the early-step window. The CV score over early reasoning steps is then given by:

$$\text{CV}(\mathcal{C}) = \frac{\sigma(\mathcal{R}_{\text{score}}^{\text{early}})}{\mu(\mathcal{R}_{\text{score}}^{\text{early}})}, \tag{3}$$

where $\mu(\cdot)$ and $\sigma(\cdot)$ denote the mean and standard deviation, respectively.

To assess **Pattern #2**, we introduce a **Attention Score** that quantifies the extent to which later reasoning steps attend to earlier shallow-pattern matching or overthinking steps (Figure 1). Let the full reasoning trace be $\mathcal{C} = \langle c_1, c_2, \dots, c_K \rangle$, and define the later reasoning steps as $\mathcal{C}_{\text{later}} = \{c_k\}_{k=\lceil \eta K \rceil}^K$. For a step $c_k \in \mathcal{C}_{\text{later}}$, we compute the mean attention from $c_k$ to each earlier step $c_j$ as:

$$\bar{a}_{k \to j} = \frac{1}{|c_k||c_j|} \sum_{t \in c_k} \sum_{s \in c_j} \left( \frac{1}{|\mathcal{L}|} \sum_{l \in \mathcal{L}} \frac{1}{H} \sum_{h=1}^H a_{t,s}^{l,h} \right),$$

where $a_{t,s}^{l,h}$ denotes the attention weight from token $t$ to token $s$ at head $h$ in layer $l$, $H$ is the number of heads per layer, $\mathcal{L}$ is the set of selected layers for aggregation, and the constant $\eta$ defines late steps.

We then identify the top-$K$ most attended earlier steps based on $\bar{a}_{k \to j}$: $\mathcal{T}_k = \text{TopK}\left( \{\bar{a}_{k \to j}\}_{j=1}^{k-1}, K \right)$, where $\mathcal{T}_k$ is the set of indices corresponding to the top-attended steps. The step-level attention score for $c_k$ is then defined as the proportion of these steps whose Reasoning Scores fall outside the normal range, either in the lower quartile or exceeding a high threshold $\tau$:

$$\text{AttnScore}(c_k) = \frac{1}{K} \sum_{j \in \mathcal{T}_k} \mathbb{1}_{\left( R_{\text{score}}^j \leq \text{Quantile}_{1/4}(\mathcal{R}_{\text{score}}) \ \text{or} \ R_{\text{score}}^j \geq \tau \right)},$$

where $\mathbb{1}_{(.)}$ is the indicator function, $\text{Quantile}_{1/4}(\mathcal{R}_{\text{score}})$ denotes the first quartile of the reasoning scores (i.e., potentially shallow pattern-matching steps), and $\tau$ is a threshold identifying potentially overthinking steps.

The trace-level attention score is computed by averaging over all later steps:

$$\text{AttnScore}(\mathcal{C}) = \frac{1}{|\mathcal{C}_{\text{later}}|} \sum_{c_k \in \mathcal{C}_{\text{later}}} \text{AttnScore}(c_k), \tag{4}$$

which reflects the extent to which later reasoning steps attend to earlier incorrect steps.

**Results.** As shown in Figure 3(b) and (c) and Appendix H, across all three domains, hallucinated reasoning traces consistently yield significantly higher CV scores and Attention scores than truthful traces. This confirms that hallucinated traces are more fluctuating in reasoning depth (Pattern #1) and more likely to attend prior incorrect steps (Pattern #2), demonstrating the generalizability of both patterns beyond the initial case study (Section 3.2.1). Detailed settings are shown in Appendix H.

### 3.3 ANALYZING THE MECHANISMS BEHIND REASONING FLUCTUATION

We investigate the underlying mechanism behind **Pattern #1**, where hallucinated reasoning traces exhibit large fluctuations in reasoning depth. Building on our case study in Section 3.2.1, we hypothesize this stems from a built-in self-verification mechanism. Key questions still include: **Q1**: What triggers verification behavior in LRMs? **Q2**: Do excessively high reasoning scores reliably signal overthinking? **Q3**: If Q2 holds, what factors lead to the emergence of such overthinking steps?

To explore these, we construct step triples $(c_1, c_2, c_3)$ from reasoning traces: (1) **Stable** triples with minimal score variation from truthful traces; (2) **Rising-1** triples from hallucinated traces with a moderate score spike ($R_{\text{score}}(c_3) < \tau$), potentially triggered by shallow pattern-matching in $c_2$; and (3) **Rising-2** triples with extreme score spikes ($R_{\text{score}}(c_3) > \tau$), to probe overthinking behaviors.

**Analysis.** For **Q1**, we compare the logical consistency between $c_1$ and $c_2$ in Rising vs. Stable triples using GPT-4o judgments. As shown in Figure 4(a), stable triples show significantly higher consistency, suggesting that verification is more likely to be triggered when earlier steps are inconsistent.

Regarding **Q2**, we assess the accuracy of $c_2$ and $c_3$ in Rising-2 triples. Figure 4(b) shows that while $c_2$ is often correct, $c_3$ introduces errors, confirming that excessively high reasoning scores reliably signal overthinking. Prompts of **Q1** and **Q2** are shown in Appendix G.

To investigate **Q3**, we firstly analyze the correlation between reasoning depth and perplexity. As shown in Figure 4(c), reasoning steps with higher $R_{\text{score}}$ generally exhibit lower perplexity, indicating more certainty outputs. However, Figure 4(d) reveals that in Rising-2 triples, $c_3$ steps, despite higher reasoning scores, have higher perplexity than those in stable triples, suggesting that overthinking may produce internally unstable generations. We term this phenomenon *spurious verification*, where the model performs misguided validation driven by outcome-based reward optimization. This insight leads us to identify a new hallucination pattern: **Pattern #3:** Overthinking steps exhibit a positive correlation between $R_{\text{score}}$ and perplexity. More details analysis are provided in Appendix F.

## 4 METHODS

### 4.1 REASONING HALLUCINATION DETECTION

Building upon the patterns uncovered in our empirical study, we propose the **R**easoning **H**allucination **D**etection algorithm (**RHD**). Our approach leverages the step-level Reasoning Score $R_{\text{score}}$ to quantify thinking depth throughout the reasoning trace, and incorporates three identified indicators of hallucination: (1) Pattern #1: large fluctuations in reasoning scores during early steps, (2) Pattern #2: incorrect backtracking to earlier shallow or overthinking steps in later stages, and (3) Pattern #3: overthinking behavior where $R_{\text{score}}$ and perplexity exhibit a positive correlation.

Given a question $Q$ and its reasoning trace $\mathcal{C}$ with step-level scores $\mathcal{R}_{\text{score}}$, we define the overall Reasoning Hallucination Score as:

$$\mathcal{H}_{\mathcal{C}} = \underbrace{\alpha_1 \cdot \text{Avg}(\mathcal{R}_{\text{score}})}_{\text{Overall Reasoning Depth}} + \underbrace{\alpha_2 \cdot \text{CV}(\mathcal{C})}_{\text{Pattern \#1}} + \underbrace{\alpha_3 \cdot \text{AttnScore}(\mathcal{C})}_{\text{Pattern \#2}} + \underbrace{\alpha_4 \cdot \text{PCC}(\mathcal{R}_{\text{score}}, \text{PPL}(\mathcal{C}))}_{\text{Pattern \#3}}, \quad (5)$$

where $\alpha_1, \alpha_2, \alpha_3, \alpha_4$ are regression coefficients. $\text{Avg}$ denotes the average reasoning score, $\text{CV}$ (Eq. 3) measures fluctuations during early-steps, $\text{AttnScore}$ (Eq. 4) captures attention on earlier hallucinated steps, and $\text{PCC}$ refers to the Pearson correlation coefficient between reasoning scores and step-level perplexity $\text{PPL}(\mathcal{C})$, computed according to Eq. 11.

### 4.2 MITIGATING HALLUCINATIONS VIA STEP-LEVEL REASONING SCORE SHAPING

Reasoning hallucinations often stem from two types of flawed steps: (1) shallow pattern-matching, reflecting shortcut behaviors, and (2) overthinking, induced by excessive and misguided verification. A core factor is outcome-based RL, which only rewards the final answer and neglects intermediate steps (Chen et al., 2025b; Valmeekam et al.; Transluce Research, 2024; Kalai et al., 2025), encouraging reward-hacking heuristics that may propagate through distillation (Wang et al., 2025).

To address this, we introduce an auxiliary process-level reward based on the **reasoning score** $R_{\text{score}}$ from Section 3.1, which measures the reasoning depth at each step. This encourages meaningful

reasoning while penalizing shallow or overthinking steps. We model the reasoning process as a finite-horizon MDP $(\mathcal{S}, \mathcal{A}, P, r, \gamma)$, where $s_t \in \mathcal{S}$ is the reasoning state at step $t$, $a_t \in \mathcal{A}$ denotes the next reasoning step, $P$ is the transition probability and $r_t$ is the reward:

$$r_t = \begin{cases} 0, & t < T, \\ R_{\text{final}}, & t = T. \end{cases}$$

**Reward Shaping with Reasoning Score.** We apply potential-based reward shaping (Ng et al., 1999):

$$\bar{r}_t = r_t + \gamma \Phi(s_{t+1}) - \Phi(s_t), \quad \text{with } \Phi(s_T) = 0,$$

which preserves the optimal policy while redistributing credit: $V'(s_t) = V(s_t) - \Phi(s_t)$, where $V(s_t) = \mathbb{E}_\pi \left[ \sum_{k=t}^{T} \gamma^{k-t} r_k \,\middle|\, s_t \right]$ is the value function of original reward and $V'(s_t)$ is the shaped.

**Potential Function Design.** To avoid encouraging overthinking, we clip the reasoning score:

$$\tilde{R}_{\text{score}}(s_t) = \begin{cases} \alpha \cdot R_{\text{score}}(s_t), & R_{\text{score}}(s_t) \leq \tau, \\ 0, & \text{otherwise,} \end{cases} \quad \Phi(s_t) = -\tilde{R}_{\text{score}}(s_t),$$

where $\alpha > 0$ and $\tau$ control the weighting strength and the threshold for overthinking, respectively.

To understand the generalization benefit of our proposed reasoning score–based shaping, we derive a uniform convergence bound under augmented rewards:

**Theorem 1** (Generalization Gap with Augmented Rewards). *Let the policy class $\Pi$ be such that for any $\pi \in \Pi$, the augmented return $R(\pi, \xi) = \sum_{t=1}^{T} \gamma^{t-1} \bar{r}_t(\xi)$ is uniformly bounded in $[0, \bar{R}_{\max}]$ for any trajectory $\xi$ sampled from the environment. Each trajectory $\xi = (s_1, a_1, \bar{r}_1, \ldots, s_T, a_T, \bar{r}_T)$ denotes a complete multi-step reasoning trace. Suppose that $\Pi$ has Rademacher complexity $\mathcal{R}_n(\Pi)$ based on $n$ independent training samples $\{\xi_i\}_{i=1}^{n}$. Then, with probability at least $1 - \delta$, for any $\pi \in \Pi$ the following holds: $J_{test}(\pi) - J_{train}(\pi) \leq 2\bar{R}_{\max} \mathcal{R}_n(\Pi) + \bar{R}_{\max} \sqrt{\frac{\log(1/\delta)}{2n}}$, where $J_{test}(\pi) = \mathbb{E}_\xi[R(\pi, \xi)]$ is the expected test return and $J_{train}(\pi) = \frac{1}{n} \sum_{i=1}^{n} R(\pi, \xi_i)$ is the empirical training return.*

The proof is given in Appendix B. Intuitively, our reasoning score acts as a regularizer that encourages logically consistent behaviors and effectively reduces the Rademacher complexity $\mathcal{R}_n(\Pi)$, thereby tightening the bound and improving generalization to unseen reasoning tasks.

**Integrate into GRPO.** To demonstrate compatibility with standard RL algorithms, we integrate the reasoning score shaping framework into the Group Relative Policy Optimization (GRPO), a scalable and widely used RL algorithm for reasoning model training DeepSeek-AI (2025); Shao et al. (2024); Dai et al. (2026), yielding **GRPO-R**. All implementation and formulation details of GRPO-R are provided in Appendix C.

## 5 EXPERIMENTS

### 5.1 REASONING HALLUCINATION DETECTION

**Data and Evaluation.** We evaluate our RHD method on the **ReTruthQA** dataset spanning three reasoning domains: `Math`, `Science`, and `MultiHopQA` (construction details in Appendix D). We adopt two evaluation settings: (1) **Binary Detection**, which assesses the model's ability to detect hallucinations in individual $(Q, C)$ pairs using AUC and PCC; (2) **Multi-Trace Ranking**, which evaluates whether the model can rank truthful traces higher among multiple candidates $(Q, \{C_1, ..., C_N\})$, following TruthfulQA-MC (Lin et al., 2021). We report MC1, MC2, and MC3 to measure hallucination ranking accuracy (Evaluation details are in Appendix I).

**Models and Baselines.** We conduct experiments on two open-source LRM: `DeepSeek-R1-Distill-Qwen-7B` (R1-7B) and `DeepSeek-R1-Distill-Qwen-14B` (R1-14B) DeepSeek-AI (2025). We compare our method against six categories of hallucination detection baselines: (1) Ensemble based self-evaluation (e.g., ChainPoll (Friel &

**Table 1:** Performance comparisons between RHD and baselines for Reasoning Hallucination Detection. The boldface represents the best performance, and the underline represents the second-best. † means improvements are significant (paired t-test or DeLong test at $p$-value $< 0.05$).

| LRMs | Categories | Methods | ReTruthQA (MATH) Binary Detection AUC | PCC | Multi-Trace Ranking MC1 | MC2 | MC3 | ReTruthQA (Science) Binary Detection AUC | PCC | Multi-Trace Ranking MC1 | MC2 | MC3 | ReTruthQA (MultiHopQA) Binary Detection AUC | PCC | Multi-Trace Ranking MC1 | MC2 | MC3 |
|---|---|---|---|---|---|---|---|---|---|---|---|---|---|---|---|---|---|
| R1-7B | Ensemble | ChainPoll | 0.6384 | 0.2603 | 0.3020 | 0.2952 | 0.3583 | 0.6468 | 0.2612 | 0.2700 | 0.2580 | 0.3098 | 0.6297 | 0.2233 | 0.4208 | 0.3019 | 0.3954 |
| | | LMvLM | 0.6364 | 0.3728 | 0.3204 | 0.2504 | 0.3402 | 0.5345 | 0.1890 | 0.2600 | 0.2100 | 0.3113 | 0.6331 | 0.2759 | 0.3649 | 0.3049 | 0.3984 |
| | | SelfCheckGPT | 0.7727 | 0.4598 | 0.4091 | 0.2784 | 0.4119 | 0.6819 | 0.2669 | 0.3793 | 0.3655 | 0.5320 | 0.6886 | 0.2955 | 0.2553 | 0.1915 | 0.3118 |
| | Uncertainty | P(True) | 0.7216 | 0.2681 | 0.5455 | 0.4068 | 0.5182 | 0.6207 | 0.2572 | 0.5172 | 0.4276 | 0.5533 | 0.5400 | 0.1684 | 0.4026 | 0.3030 | 0.4032 |
| | | LN-Entropy | 0.6896 | 0.3099 | 0.5000 | 0.3917 | 0.5096 | 0.5553 | 0.1129 | 0.3700 | 0.3200 | 0.3113 | 0.6123 | 0.2149 | 0.4156 | 0.3208 | 0.4461 |
| | | PPL | 0.7025 | 0.2856 | 0.5909 | 0.4205 | 0.5267 | 0.5434 | 0.1144 | 0.3793 | 0.3034 | 0.3990 | 0.6432 | 0.2249 | 0.5745 | 0.4532 | 0.5241 |
| | Length | Length-Score | 0.5351 | 0.0922 | 0.4318 | 0.2568 | 0.3408 | 0.5510 | 0.0911 | 0.5793 | 0.5034 | 0.5737 | 0.5815 | 0.1496 | 0.5106 | 0.3887 | 0.4674 |
| | PRM | Qwen2.5-PRM800K | 0.6601 | 0.2746 | 0.4773 | 0.3000 | 0.4572 | 0.6153 | 0.2203 | 0.4400 | 0.3605 | 0.4444 | 0.5694 | 0.1074 | 0.5065 | 0.4167 | 0.4990 |
| | | Qwen2.5-PRM-7B | 0.5563 | 0.1354 | 0.4318 | 0.2701 | 0.3913 | 0.5690 | 0.1275 | 0.2200 | 0.1425 | 0.2382 | 0.5422 | 0.0866 | 0.4026 | 0.2952 | 0.3947 |
| | LCM | GPT4-o | 0.7513 | 0.3794 | 0.4091 | 0.2705 | 0.4131 | 0.7045 | 0.2026 | 0.2500 | 0.2965 | 0.3200 | 0.7123 | 0.2204 | 0.4043 | 0.2830 | 0.3704 |
| | | Qwen2.5-32B | 0.6942 | 0.2082 | 0.2500 | 0.1955 | 0.2935 | 0.6525 | 0.2635 | 0.3103 | 0.2897 | 0.4458 | 0.6424 | 0.2056 | 0.4400 | 0.3300 | 0.4187 |
| | Self-Aware | UQAC | 0.6671 | 0.2902 | 0.5833 | 0.3715 | 0.5298 | 0.6303 | 0.2369 | 0.4700 | 0.3925 | 0.4885 | 0.6736 | 0.2583 | 0.6623 | 0.5335 | 0.6425 |
| | | EigenScore | 0.7539 | 0.3868 | 0.4583 | 0.3250 | 0.3007 | 0.6488 | 0.2601 | 0.4260 | 0.3777 | 0.3815 | 0.6696 | 0.2858 | 0.5195 | 0.4113 | 0.3885 |
| | Ours | **RHD** | **0.7978** | **0.4852**† | **0.6591**† | **0.4765**† | **0.5699**† | **0.7194** | **0.3060**† | **0.6207**† | **0.5448**† | **0.6009**† | **0.7361**† | **0.3863**† | **0.7660**† | **0.6255**† | **0.7103**† |
| R1-14B | Ensemble | ChainPoll | 0.5858 | 0.1658 | 0.2704 | 0.2535 | 0.3394 | 0.6640 | 0.3134 | 0.3261 | 0.1775 | 0.2188 | 0.5846 | 0.1607 | 0.2319 | 0.1972 | 0.2638 |
| | | LMvLM | 0.6620 | **0.3835** | 0.2563 | 0.2507 | 0.3133 | 0.5435 | 0.2132 | 0.3333 | 0.2300 | 0.3421 | 0.6250 | 0.2914 | 0.2042 | 0.1885 | 0.2506 |
| | | SelfCheckGPT | 0.5823 | 0.2923 | 0.2462 | 0.2167 | 0.2930 | 0.5109 | 0.1048 | 0.3287 | 0.2566 | 0.3683 | 0.5208 | 0.1268 | 0.3167 | 0.3083 | 0.0320 |
| | Uncertainty | P(True) | 0.6460 | 0.1443 | 0.2615 | 0.2374 | 0.4570 | 0.6645 | 0.2582 | 0.4828 | 0.3460 | 0.4885 | 0.6090 | 0.2057 | 0.3147 | 0.2508 | 0.4107 |
| | | LN-Entropy | 0.6423 | 0.2242 | 0.3479 | 0.2939 | 0.4754 | 0.6248 | 0.2134 | 0.5862 | 0.4147 | 0.5264 | 0.5337 | 0.0494 | 0.3125 | 0.2340 | 0.3678 |
| | | PPL | 0.6526 | 0.2330 | **0.3846** | 0.2744 | 0.4444 | 0.6219 | 0.1182 | 0.6000 | 0.4215 | 0.5162 | 0.5337 | 0.1701 | 0.3058 | 0.2521 | 0.3630 |
| | Length | Length-Score | 0.5184 | 0.0810 | 0.2817 | 0.2329 | 0.3400 | 0.5814 | 0.1487 | 0.5345 | 0.3848 | 0.4211 | 0.5971 | 0.1843 | 0.4711 | 0.3434 | 0.4284 |
| | PRM | Qwen2.5-PRM800K | 0.5708 | 0.1285 | 0.3077 | 0.2697 | 0.4028 | 0.7267 | 0.4100 | 0.5862 | 0.3819 | 0.5132 | 0.6579 | 0.2451 | 0.4476 | 0.3366 | 0.4702 |
| | | Qwen2.5-PRM-7B | 0.5416 | 0.1249 | 0.3538 | 0.2918 | 0.4429 | 0.6983 | 0.3633 | 0.6133 | 0.4556 | 0.5449 | 0.6274 | 0.2758 | 0.5045 | 0.3642 | 0.4853 |
| | LCM | GPT4-o | 0.6604 | 0.2458 | 0.2154 | 0.1785 | 0.3073 | 0.6265 | 0.1344 | 0.3333 | 0.1628 | 0.1933 | 0.6328 | 0.2356 | 0.2517 | 0.1878 | 0.2683 |
| | | Qwen2.5-32B | 0.6650 | 0.3055 | 0.2676 | 0.2451 | 0.3632 | 0.6974 | 0.2381 | 0.3833 | 0.2150 | 0.3428 | 0.7071 | 0.2716 | 0.3472 | 0.2517 | 0.4177 |
| | Self-Aware | UQAC | 0.6374 | 0.2303 | 0.3444 | 0.2836 | **0.5104** | 0.7157 | 0.3732 | 0.6207 | 0.4170 | 0.4885 | 0.6952 | 0.3397 | 0.5417 | 0.4222 | 0.4988 |
| | | EigenScore | 0.6706 | 0.3496 | 0.3282 | 0.2282 | 0.3388 | 0.6146 | 0.2228 | 0.4623 | 0.3643 | 0.3547 | 0.6719 | 0.3056 | 0.3694 | 0.3542 | 0.3750 |
| | Ours | **RHD** | **0.7292**† | 0.3476 | 0.3692 | **0.3005** | 0.4644 | **0.7686**† | **0.4625**† | **0.6667**† | **0.4714**† | **0.5671**† | **0.7255**† | **0.3742**† | **0.5785**† | **0.4421**† | **0.5154**† |

**Table 2:** Performance comparisons between GRPO-R and baselines. Bold indicates the best result.

| Models | DeepSeek-R1-1.5B MATH500 | AIME(2024) | GPQA(diamond) | GPQA(main) | GPQA(extended) | Qwen2.5-1.5B-Instruct MATH500 | AIME(2024) | GPQA(diamond) | GPQA(main) | GPQA(extended) |
|---|---|---|---|---|---|---|---|---|---|---|
| Base | 0.772 | 0.333 | 0.354 | 0.333 | 0.339 | 0.466 | 0.100 | 0.202 | 0.197 | 0.211 |
| +GRPO | 0.770 | 0.333 | 0.359 | 0.335 | **0.359** | 0.480 | 0.033 | **0.247** | 0.214 | 0.266 |
| +GRPO-R | **0.788** | **0.367** | **0.414** | **0.371** | 0.357 | **0.490** | **0.133** | **0.247** | **0.243** | **0.275** |

Sanyal, 2023)); (2) Uncertainty based methods (e.g., P(True) Kadavath et al. (2022)); (3) Self-Awareness based approaches (e.g., UQAC Li et al. (2025b)); (4) LLM-as-Critic (LCM) models (e.g., `GPT-4o`); (5) Process Reward Models (PRMs) with step-level supervision (e.g., `Qwen2.5-Math-PRM`); (6) Length-based scoring, which uses trace length as a proxy for hallucination likelihood. Baselines and RHD implementation details are in Appendix I and J.

**Main Results.** As shown in Table 1, RHD consistently outperforms most baselines across all ReTruthQA domains, model backbones, and evaluation settings, demonstrating strong robustness. Ensemble and LCM methods perform well in binary detection but struggle in multi-trace ranking, indicating difficulty in fine-grained comparison. Uncertainty-based methods are sensitive to output length, while Process Reward Models often suffer from limited generalization. In contrast, RHD directly leverages reasoning mechanisms for more accurate detection. Self-awareness methods perform competitively but lack explicit reasoning analysis. The Length-based baseline performs well in multi-trace settings—supporting the intuition that overly long traces are more error-prone, but underperforms in binary detection, limiting its generality. These findings highlight the effectiveness of RHD modeling internal reasoning patterns for hallucination detection. Additional Qwen3-8B Yang et al. (2025) results, ablations and sensitivity studies are provided in Appendix L, M, N, and K.

## 5.2 REASONING HALLUCINATION MITIGATION

**Experimental Setting.** To assess the effectiveness of GRPO-R in reducing reasoning hallucinations, we fine-tune `Qwen2.5-1.5B-Instruct` and `DeepSeek-R1-1.5B` on 2,000 examples from `OpenR1-Math-220K` (Team, 2024) using either GRPO or our proposed GRPO-R. We evaluate the accuracy (Hugging Face, 2025) on two in-domain math benchmarks—`MATH500` Lightman et al. (2023) and `AIME 2024` AI-MO (2024a)—and an out-of-distribution science benchmark—`GPQA` (Rein et al., 2024). Implementation details are in Appendix O.

**Main Results.** As shown in Table 2, GRPO-R outperforms GRPO across most of the tasks, indicating that shaping reasoning steps via the reasoning score enhances both factual accuracy and reasoning reliability. Gains on `GPQA` further suggest improved generalization beyond training distribution. Additional sensitive and GRPO-variants analyses are in Appendix P and R. Hallucination mitigation experiments in data distillation in Appendix Q further validate the effectiveness of RHD model.

## 6 CONCLUSION

We study *Reasoning Hallucination* in LRMs from a mechanistic perspective, probing internal model behaviors rather than surface text. We propose the **Reasoning Score**, a step-level metric grounded in mechanistic interpretability that quantifies reasoning depth. Using this lens, we uncover three characteristic hallucination patterns—early-stage depth fluctuations, incorrect backtracking, and spurious verification-induced overthinking—and build the **RHD** framework for their detection. Finally, we introduce **GRPO-R**, which integrates reasoning-score–based shaping into reinforcement learning, improving accuracy and robustness across reasoning benchmarks. This establishes a unified pipeline from mechanistic analysis to practical mitigation of reasoning hallucinations.

## ACKNOWLEDGMENTS

This work was funded by the National Natural Science Foundation of China (62472426, 62376275), fund for building world-class universities (disciplines) of Renmin University of China. Supported by the School of Interdisciplinary Studies, Renmin University of China, and the Big Data and Responsible Artificial Intelligence for National Governance (BRAIN), Renmin University of China. It constitutes a phased research output of the Young Researcher Research Fund of the Renmin University of China–Westlake University Joint Academy on Future Humanity. Supported by the Outstanding Innovative Talents Cultivation Funded Programs 2026 of Renmin University of China. Work partially done at Engineering Research Center of Next-Generation Intelligent Search and Recommendation, Ministry of Education, and Beijing Key Laboratory of Research on Large Models and Intelligent Governance.

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

CONTENTS

## A USE OF LARGE LANGUAGE MODELS

In accordance with the ICLR 2026 policy on the disclosure of language model usage, we confirm that Large Language Models (LLMs) were utilized in the preparation of this paper. The usage was limited to aiding with language fluency, grammar checking, and polishing of the writing. The research ideas, experimental design, theoretical analysis, and all scientific contributions were solely developed by the authors. No LLMs contributed at the level of a contributing author.

**Disclosure:** Yes, to aid or polish writing. Details are described in the paper.

## B PROOF OF GENERALIZATION GAP WITH AUGMENTED REWARDS

*Proof of Theorem 1.* For any policy $\pi \in \Pi$, define the augmented return

$$R(\pi, \xi) = \sum_{t=1}^{T} \gamma^{t-1} \bar{r}_t(\xi).$$

Assume that $\bar{r}_t(\xi) \in [0, \bar{r}_{\max}]$ for each $t$, so that

$$R(\pi, \xi) \in [0, \bar{R}_{\max}].$$

Define the expected return:

$$J_{\text{test}}(\pi) = \mathbb{E}_{\xi \sim \mathcal{D}} \left[ R(\pi, \xi) \right],$$

and the empirical return:

$$J_{\text{train}}(\pi) = \frac{1}{n} \sum_{i=1}^{n} R(\pi, \xi_i).$$

We aim to bound the expected generalization gap between the test return and empirical return for policies in class $\Pi$ via Rademacher complexity. Let the function class be defined as

$$\mathcal{F} = \{ f_\pi(\xi) = R(\pi, \xi) \mid \pi \in \Pi \},$$

where $R(\pi, \xi)$ is the total return over trajectory $\xi$ under policy $\pi$ using the augmented reward $\bar{r}_t$. Our goal is to bound:

$$\sup_{\pi \in \Pi} |J_{\text{test}}(\pi) - J_{\text{train}}(\pi)| = \sup_{f \in \mathcal{F}} \left| \mathbb{E}[f(\xi)] - \frac{1}{n} \sum_{i=1}^{n} f(\xi_i) \right|.$$

Let $\xi_1, \ldots, \xi_n$ be the training samples drawn i.i.d. from the environment distribution $\mathcal{D}$, and $\xi_1', \ldots, \xi_n'$ be another independent copy drawn from the same distribution. By using an independent ghost sample set and the triangle inequality, we have:

$$\mathbb{E}_{\{\xi_i\}} \left[ \sup_{f \in \mathcal{F}} \left( \mathbb{E}_{\xi \sim \mathcal{D}}[f(\xi)] - \frac{1}{n} \sum_{i=1}^{n} f(\xi_i) \right) \right] = \mathbb{E}_{\{\xi_i\}, \{\xi_i'\}} \left[ \sup_{f \in \mathcal{F}} \left( \frac{1}{n} \sum_{i=1}^{n} f(\xi_i') - f(\xi_i) \right) \right]$$

$$\leq \mathbb{E}_{\{\xi_i\}, \{\xi_i'\}} \left[ \sup_{f \in \mathcal{F}} \frac{1}{n} \sum_{i=1}^{n} (f(\xi_i') - f(\xi_i)) \right].$$

To simplify the expression, we now introduce independent Rademacher variables $\sigma_1, \ldots, \sigma_n \in \{-1, +1\}$, where each $\sigma_i$ takes value $+1$ or $-1$ with equal probability. Since $f(\xi_i') - f(\xi_i)$ is symmetric around zero due to $\xi_i \sim \xi_i'$, we can write:

$$\mathbb{E}_{\{\xi_i\}, \{\xi_i'\}} \left[ \sup_{f \in \mathcal{F}} \frac{1}{n} \sum_{i=1}^{n} (f(\xi_i') - f(\xi_i)) \right] = \mathbb{E}_{\{\xi_i\}, \{\xi_i'\}, \{\sigma_i\}} \left[ \sup_{f \in \mathcal{F}} \frac{1}{n} \sum_{i=1}^{n} \sigma_i (f(\xi_i') - f(\xi_i)) \right].$$

We now apply the triangle inequality again:

$$\sup_{f \in \mathcal{F}} \sum_{i=1}^{n} \sigma_i (f(\xi_i') - f(\xi_i)) \leq \sup_{f \in \mathcal{F}} \sum_{i=1}^{n} \sigma_i f(\xi_i') + \sup_{f \in \mathcal{F}} \sum_{i=1}^{n} (-\sigma_i) f(\xi_i).$$

Since $-\sigma_i$ is still a Rademacher variable and $\xi_i$ and $\xi_i'$ have the same distribution, the two expectations are equal. Thus, we obtain:

$$\mathbb{E}_{\{\xi_i\},\{\xi_i'\}}\left[\sup_{f\in\mathcal{F}}\frac{1}{n}\sum_{i=1}^{n}\left(f(\xi_i')-f(\xi_i)\right)\right] \leq 2\mathbb{E}_{\{\xi_i\},\{\sigma_i\}}\left[\sup_{f\in\mathcal{F}}\frac{1}{n}\sum_{i=1}^{n}\sigma_i f(\xi_i)\right]$$
$$= 2\mathcal{R}_n(\mathcal{F}),$$

where $\mathcal{R}_n(\mathcal{F})$ is the empirical Rademacher complexity of $\mathcal{F}$.

Assume every return is bounded, $0 \leq f_\pi(\xi) \leq \bar{R}_{\max}$, and that $f_\pi(\xi)$ is linear in the augmented per–step rewards $\bar{r}_t(\xi)$:

$$f_\pi(\xi) = \sum_{t=1}^{T}\gamma^{t-1}\bar{r}_t(\xi).$$

Introduce the normalised return $\tilde{f}_\pi(\xi) := f_\pi(\xi)/\bar{R}_{\max} \in [0,1]$ and let $\tilde{\mathcal{F}} := \{\tilde{f}_\pi \mid \pi \in \Pi\}$. Because Rademacher complexity is positively homogeneous in its function class,

$$\mathcal{R}_n(\mathcal{F}) = \mathcal{R}_n\big(\bar{R}_{\max}\tilde{\mathcal{F}}\big) = \bar{R}_{\max}\mathcal{R}_n(\tilde{\mathcal{F}}).$$

We measure the complexity of the policy class precisely through these normalised returns and set

$$\mathcal{R}_n(\Pi) := \mathcal{R}_n(\tilde{\mathcal{F}}).$$

*Justification.* Even if the mapping $\pi \mapsto \tilde{f}_\pi$ is not injective, Rademacher complexity is **monotone** with respect to set inclusion: enlarging the function class can only increase $\mathcal{R}_n$. Hence analysing the (possibly larger) class $\tilde{\mathcal{F}}$ yields a conservative upper bound on the true policy complexity—exactly what we need for a valid generalisation bound.

Combining the two displays yields

$$\mathcal{R}_n(\mathcal{F}) \leq \bar{R}_{\max}\mathcal{R}_n(\Pi)$$

(the identity can be written as "$\leq$" because any alternative normalisation would only shrink the right–hand side).

Substituting the above bound into the symmetrisation result, we obtain

$$\mathbb{E}\left[\sup_{\pi\in\Pi}\big|J_{\text{test}}(\pi)-J_{\text{train}}(\pi)\big|\right] \leq 2\bar{R}_{\max}\mathcal{R}_n(\Pi),$$

We now move from the expected generalization gap to a high-probability bound that holds uniformly over all policies $\pi \in \Pi$.

Let $X_i = R(\pi,\xi_i) = \sum_{t=1}^{T}\gamma^{t-1}\bar{r}_t(\xi_i)$ be the augmented return of policy $\pi$ on the $i$-th training trajectory. Then $J_{\text{train}}(\pi) = \frac{1}{n}\sum_{i=1}^{n}X_i$ and $J_{\text{test}}(\pi) = \mathbb{E}_{\xi\sim\mathcal{D}}[X_i]$. By assumption, $X_i \in [0,\bar{R}_{\max}]$.

Applying Hoeffding's inequality for bounded i.i.d. variables, we have for any fixed $\pi \in \Pi$:

$$\Pr\left(|J_{\text{test}}(\pi)-J_{\text{train}}(\pi)| \geq \varepsilon\right) \leq 2\exp\left(-\frac{2n\varepsilon^2}{(\bar{R}_{\max})^2}\right).$$

Solving for $\varepsilon$ yields that with probability at least $1-\delta$,

$$|J_{\text{test}}(\pi)-J_{\text{train}}(\pi)| \leq \bar{R}_{\max}\sqrt{\frac{\log(1/\delta)}{2n}}. \tag{16}$$

Define the worst-case generalization gap over the policy class:

$$\Delta(\mathcal{S}) := \sup_{\pi\in\Pi}\left(J_{\text{test}}(\pi)-J_{\text{train}}(\pi)\right),$$

where $\mathcal{S} = \{\xi_1,\dots,\xi_n\}$ is the training set.

*(i) Expected bound from above:* Using symmetrization and Rademacher complexity arguments, we already established:

$$\mathbb{E}_{\mathcal{S}}[\Delta(\mathcal{S})] \leq 2\bar{R}_{\max}\mathcal{R}_n(\Pi). \tag{6}$$

*(ii) High-probability deviation bound via McDiarmid's inequality:* Let us show that $\Delta(\mathcal{S})$ concentrates around its expectation. Consider replacing any single sample $\xi_i$ in $\mathcal{S}$ by an independent copy $\xi_i'$. Because each return $X_i = R(\pi, \xi_i)$ is bounded in $[0, \bar{R}_{\max}]$ and each contributes $\frac{1}{n}$ to the empirical mean, the influence of changing $\xi_i$ is bounded by:

$$\left|\Delta(\mathcal{S}) - \Delta(\mathcal{S}^{(i)})\right| \leq \frac{\bar{R}_{\max}}{n}.$$

Hence, $\Delta(\mathcal{S})$ is $\bar{R}_{\max}/n$-Lipschitz in each of its $n$ arguments.

Applying McDiarmid's inequality:

$$\Pr\left(\Delta(\mathcal{S}) - \mathbb{E}[\Delta(\mathcal{S})] \geq \varepsilon\right) \leq \exp\left(-\frac{2\varepsilon^2}{\sum_{i=1}^{n}(\bar{R}_{\max}/n)^2}\right) = \exp\left(-\frac{2n\varepsilon^2}{(\bar{R}_{\max})^2}\right).$$

Solving for $\varepsilon$ again yields that with probability at least $1 - \delta$,

$$\Delta(\mathcal{S}) \leq \mathbb{E}[\Delta(\mathcal{S})] + \bar{R}_{\max}\sqrt{\frac{\log(1/\delta)}{2n}}. \tag{7}$$

*(iii) Final generalization gap:* Combining Equation 6 and 7, with probability at least $1 - \delta$ over the random draw of the training set $\mathcal{S}$, we obtain:

$$\sup_{\pi \in \Pi}\left[J_{\text{test}}(\pi) - J_{\text{train}}(\pi)\right] \leq 2\bar{R}_{\max}\mathcal{R}_n(\Pi) + \bar{R}_{\max}\sqrt{\frac{\log(1/\delta)}{2n}}.$$

Equivalently, for all $\pi \in \Pi$,

$$J_{\text{test}}(\pi) - J_{\text{train}}(\pi) \leq 2\bar{R}_{\max}\mathcal{R}_n(\Pi) + \bar{R}_{\max}\sqrt{\frac{\log(1/\delta)}{2n}} \tag{8}$$

$\square$

**Conclusion.** Equation 8 provides a uniform generalization gap for any policy $\pi \in \Pi$, showing that the expected test-time performance is lower bounded by the training performance minus a complexity-dependent regularization term. According to this theorem, as the augmented reward $R_{\text{score}}(s_t)$ is well-aligned with genuine logical reasoning, it acts as a regularizer that effectively reduces the Rademacher complexity $\mathcal{R}_n(\Pi)$, thereby tightening the bound. This theoretical result highlights that our proposed process supervision framework not only improves credit assignment during training but also enhances generalization to unseen reasoning tasks.

This theoretical result not only explains why our process supervision framework enhances generalization to unseen reasoning tasks, but also sheds light on the hallucination risk in outcome-based RL. Because outcome-only reward collapses trajectories with differing reasoning quality into a shared positive label, it greatly increases the functional hypothesis class and thereby the generalization gap. As a result, models trained with such reward signals are more likely to memorize spurious patterns and produce hallucinated reasoning at test time.

## C  DETAILED IMPLEMENTATION OF GRPO-R

Our proposed process-level reasoning score supervision is compatible with any token-level RL algorithm. In this work, we instantiate it within Group Relative Policy Optimization (GRPO), yielding **GRPO-R**. GRPO is a scalable and widely used RL framework for reasoning model training, which promotes the generation of high-quality reasoning trajectories by ranking $G$ candidate outputs based on their relative returns, without relying on explicit value estimation DeepSeek-AI (2025); Shao et al. (2024).

**Table 3:** Statistics of ReTruthQA dataset across domains.

| Dataset | #Samples | #Traces | Avg Truthful Traces | Avg Hallucination Traces |
|---------|----------|---------|---------------------|--------------------------|
| MATH | 57 | 417 | 3.35 | 3.96 |
| Science | 88 | 541 | 3.05 | 3.10 |
| MultiHopQA | 184 | 1186 | 2.74 | 3.70 |

Given a prompt $q$ and $G$ outputs $\{o_i\}_{i=1}^G$, each output $o_i$ corresponds to a sequence of reasoning states $\{s_{i,1}, \ldots, s_{i,K}\}$ produced over $K$ reasoning steps. In the original GRPO setup, only the final step receives a nonzero reward:

$$r_i^{\text{step}}(j) = \begin{cases} r_i^{\text{final}}, & j = K, \\ 0, & j < K, \end{cases}$$

where $r_i^{\text{final}}$ denotes the scalar reward assigned to the final outcome.

We replace this sparse signal with our shaped step-level reward using potential-based reward shaping:

$$\bar{r}_i^{\text{step}}(j) = \tilde{r}_i^{\text{step}}(j) - \gamma \tilde{R}_{\text{score}}(s_{i,j+1}) + \tilde{R}_{\text{score}}(s_{i,j}),$$

where $\tilde{R}_{\text{score}}(s) = \min\big(R_{\text{score}}(s), \tau\big)$ and we set $\gamma = 1$. These shaped rewards are collected into the set $\mathbf{R}'$, standardized as:

$$\hat{r}_i^{\text{step}}(j) = \frac{\bar{r}_i^{\text{step}}(j) - \text{mean}(\mathbf{R}')}{\text{std}(\mathbf{R}')},$$

and used to compute token-level advantages:

$$\hat{A}_{i,t} = \sum_{j:\,\text{step}(j) \geq t} \hat{r}_i^{\text{step}}(j).$$

Finally, we optimize the policy using the enhanced GRPO objective, termed GRPO-R:

$$\begin{aligned}
\mathcal{J}_{\text{GRPO-R}}(\theta) = \mathbb{E}_{q \sim P(Q),\, \{o_i\} \sim \pi_{\theta_{\text{old}}}(O|q)} &\Bigg[ \sum_{i=1}^G \sum_{t=1}^{|o_i|} \min\Big( \frac{\pi_\theta(o_{i,t} \mid q, o_{i,<t})}{\pi_{\theta_{\text{old}}}(o_{i,t} \mid q, o_{i,<t})} \hat{A}_{i,t}, \\
&\qquad \text{clip}\Big(\frac{\pi_\theta(o_{i,t} \mid q, o_{i,<t})}{\pi_{\theta_{\text{old}}}(o_{i,t} \mid q, o_{i,<t})}, 1 - \epsilon, 1 + \epsilon\Big) \hat{A}_{i,t}\Big) \\
&\qquad - \beta \cdot \text{D}_{\text{KL}}\left[\pi_\theta \| \pi_{\text{ref}}\right] \Bigg].
\end{aligned} \tag{9}$$

**Relation to Factuality-Based RL.** Factuality-oriented RL methods improve a model's alignment with external world knowledge by assigning outcome-level factual rewards (Li & Ng, 2025). These approaches address factual hallucinations and supervise only the final answer.

In contrast, GRPO-R targets reasoning hallucinations—errors originating from the model's internal multi-step reasoning process. Our method introduces a step-level reward derived from the Reasoning Score (§ 3.1), which regularizes the model's intermediate reasoning dynamics rather than factual correctness alone.

These two types of RL are complementary: factual RL enhances knowledge faithfulness, while GRPO-R improves process faithfulness by promoting deep, coherent reasoning. Empirically (Table 2), reinforcing internal reasoning also brings secondary gains in factual reliability.

# D  RETRUTHQA CONSTRUCTION

## D.1  DATA SOURCES AND MODELS

Due to the absence of dedicated datasets for evaluating reasoning hallucination detection—particularly for strong open-source LRMs such as `DeepSeek-R1-7B` and `R1-14B`, we construct a new benchmark specifically tailored to multi-step reasoning tasks following the previous annotation process of

the hallucination detection dataset (Niu et al., 2024). Unlike traditional hallucinations, *reasoning hallucinations* are often embedded within logically coherent reasoning traces, which makes the incorrect information more persuasive and substantially harder to identify. This intrinsic challenge necessitates careful and fine-grained human annotation in order to ensure reliable evaluation.

We select three major categories of reasoning tasks: `Math`, `Science`, and `MultiHopQA`.

For `Math`, we construct the dataset using benchmark datasets commonly used for evaluating mathematical reasoning capabilities, including `MATH500` (Lightman et al., 2023), `AMC 2023` (AI-MO, 2024b), and `AIME 2024` (AI-MO, 2024a).

For `Science`, we adopt `GPQA` (Rein et al., 2024), a PhD-level science multiple-choice QA dataset with questions authored by domain experts in physics, chemistry, and biology.

For `MultiHopQA`, we randomly sample 1000 questions from four multi-hop QA datasets: `HotpotQA` (Yang et al., 2018), `2WikiMultihopQA` (Ho et al., 2020), `MuSiQue` (Trivedi et al., 2022), and `Bamboogle` (Press et al., 2022).

For each question, we generate 20 responses using `DeepSeek-R1-Distill-Qwen-7B` and `DeepSeek-R1-Distill-Qwen-14B` via random sampling. The prompting format is as follows:

**Math:**

```
Please answer the following math question.
You should provide your final answer in the format \boxed{YOUR_ANSWER}.
Separate your following steps using \n\n.
Question:\n\n
```

**Science:**

```
Please answer the following multiple-choice question.
You should provide your final choice in the format \boxed{YOUR_CHOICE}.
Separate your following steps using \n\n.
Question:\n\n
```

**MultiHopQA:**

```
Please answer the following question.
You should provide your final answer in the format \boxed{YOUR_ANSWER}.
Separate your following steps using \n\n.
Question:\n\n
```

### D.2 REASONING STEP SEGMENTATION STRATEGY

We adopt a two-stage segmentation procedure. First, we split the reasoning trace based on cognitive behavior tokens such as `</think>`, `Wait`, `But`, `However`, `Hmm`, `Alternatively`, which typically mark transitions in reasoning patterns. Then, we apply a finer-grained split based on formatting: as specified in the prompt, the LRM is instructed to separate reasoning steps using `\n\n`, which we use as a delimiter. This hybrid approach ensures both rule-based and model-aligned step boundaries.

### D.3 ANNOTATION PROCESS

**1. Automatic hallucination trace identification.** To ensure precision and avoid noise caused by random model errors, a reasoning trace is labeled as hallucinated only if its rollout becomes incorrect with a failure rate exceeding 90% from a specific reasoning step onward, measured over 16 rollouts. We adopt a binary search–style trace slicing procedure inspired by OmegaProcess (Luo et al., 2024) to efficiently identify hallucination points. This strategy ensures stability and causality in hallucination step detection, avoiding incidental errors due to sampling randomness. For the Science domain,

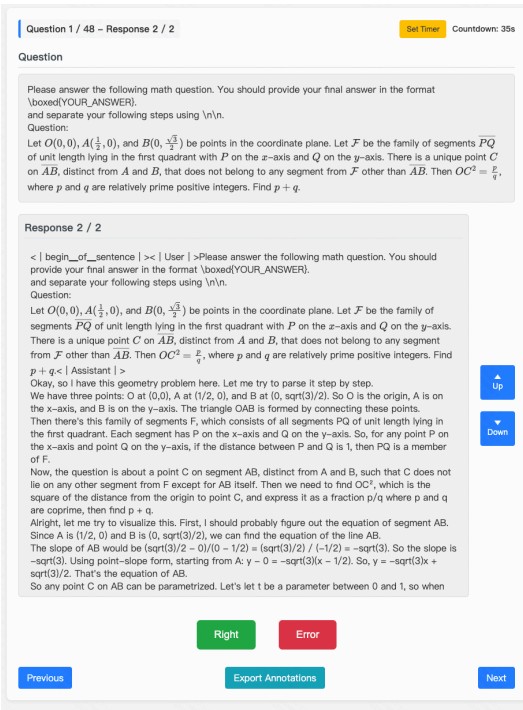

**Figure 5:** Interface Display of the Data Annotation Platform.

which mainly consists of multiple-choice questions and may contain correct guesses, we additionally perform multiple random rollouts for traces with correct answers to ensure a success rate above 90% before labeling them as truthful.

**2. Filtering non-hallucination failures.** We use GPT-4o-Mini to exclude samples where the incorrect final answer is due to clearly flawed or illogical reasoning, which does not satisfy our definition of hallucination (i.e., coherent and persuasive chains with underlying logical or factual errors). The filtering prompt is:

---

Please evaluate if the following reasoning for the given question is logically sound and leads to a correct solution.
Only respond with a score between 0 and 1, where:
0: completely incorrect or illogical reasoning
1: perfectly sound and correct reasoning

```
Question:   {question}
Reasoning:  {reasoning}
```
Score (0–1):

---

**3. Human validation.** We further perform human annotation to verify borderline cases. Two annotators with at least undergraduate-level backgrounds in computer science independently assess whether the reasoning trace is valid. We developed a web-based annotation platform with a timer (Figure 5) to standardize reading time. Based on average reading speeds (200–300 wpm for academic text), and trace lengths (typically 2000–3000 words), we set the following maximum judgment times: (1) MultiHopQA: 3 minutes (2) Math: 5 minutes (3) Science: 8 minutes

Annotators must determine within the allotted time whether a reasoning trace contains hallucinations. If they fail to identify an error in time, the trace is labeled as correct. Cases judged correct by humans but verified to be incorrect are labeled as hallucinations, ensuring that the resulting dataset captures only traces that genuinely mislead users, which is aligned with the definition of reasoning hallucination.

Final dataset statistics are shown in Table 3. For the **Multi-Trace Ranking Setting**, we directly use the collected hallucinated and truthful responses. For the **Binary Detection Setting**, which focuses on single-response accuracy, we retain one hallucinated and one truthful response per question to reflect more realistic ad-hoc usage scenarios.

## E   GSM-NoOp CONSTRUCTION PROCESS

Following the construction procedure proposed in Mirzadeh et al. (2024), we randomly sample 300 examples from the GSM8K dataset. For each question, we use `GPT-4o` to generate a No-Op phrase using the following prompt:

> Given the following math question, generate a seemingly relevant but ultimately inconsequential statement (No-Op) that can be added to the question without affecting its solution.
> Question: {Question}
> Generate a No-Op statement that:
> 1. Is short and concise
> 2. Seems relevant to the context
> 3. Does not affect the mathematical reasoning
> 4. Is natural and fits grammatically
> No-Op statement:

We then use `GPT-4o` to combine the generated No-Op phrase with the original question using the following prompt:

> Please combine the following math question and No-Op phrase into a single, natural-sounding question. The No-Op phrase should be integrated smoothly without changing the mathematical meaning.
> Math Question: {Question} No-Op Phrase: {NoOp Phrase}
> Combined Question:

The merged questions form our constructed **GSM-NoOp** dataset.

To evaluate whether the generated reasoning steps are misled by the inserted No-Op phrase, we prompt `GPT-4o` with the following instruction:

> Please evaluate if the following reasoning step is being misled by the given No-Op phrase.
> Provide a score between 0 and 1, where:
> a. 0 means the step is not misled by the No-Op phrase at all
> b. 1 means the step is completely misled by the No-Op phrase
> c. Values in between indicate partial misleading
>
> Note: Simply mentioning the No-Op phrase does not count as being misled. If the step mentions the No-Op phrase but explicitly rejects or explains why it is irrelevant to solving the problem, this should be scored as 0.
> Reasoning step: {Reasoning Step} No-Op phrase: {NoOp Phrase}
> Please provide only a number between 0 and 1, with up to 2 decimal places, wrapped in \boxed{}. For example: \boxed{0.85}

## F   DETAILS OF UNDERSTANDING THE MECHANISMS BEHIND REASONING HALLUCINATION PATTERNS

In this section, we focus on analyzing the underlying cause of **Pattern #1**, as **Pattern #2** has already been explained through the attention behavior of LRMs in the previous section. Pattern #1 highlights that hallucinated reasoning traces tend to exhibit larger fluctuations in reasoning depth, particularly in the early steps. Inspired by our preliminary analysis in § 3.2.1, we hypothesize that this may stem

from the model's built-in verification capability. However, several key questions remain: **Q1**: What triggers verification behavior in LRMs? **Q2**: Do excessively high reasoning scores genuinely indicate overthinking? **Q3**: If Q2 holds, what factors lead to the emergence of such overthinking steps?

To answer these questions, we construct reasoning step triples $(c_1, c_2, c_3)$ with different properties drawn from reasoning traces: **Stable**: The first type consists of triples from truthful traces where adjacent steps differ in $R_{\text{score}}$ by less than 0.1, representing stable reasoning. **Rising-1**: The second type contains hallucinated triples where $R_{\text{score}}(c_3) - R_{\text{score}}(c_2) > 1$ and $R_{\text{score}}(c_3) < 4$, used to analyze verification triggered by shallow pattern-matching. **Rising-2**: The third type is similar to Rising-1 but with $R_{\text{score}}(c_3) > 4$, aimed at understanding overthinking induced by verification. We construct Stable, Rising-1, and Rising-2 triples to probe dynamics (early fluctuation, over-verification). Each set contains 600 step-triples per domain (Math, Science, MultiHopQA), totaling 1,800 triples, with balanced sizes for fair comparison.

**Analysis.** To investigate **Q1**, we analyze whether reasoning steps $c_1$ and $c_2$ in the stable and rising (Rising-1 + Rising-2) triples are logically consistent, using GPT-4o as the judge (prompt details in Appendix G). As shown in Figure 4(a), the stable triples exhibit significantly higher consistency between $c_1$ and $c_2$ than rising triples, indicating that LRMs are more likely to trigger verification when early steps are internally inconsistent.

To examine **Q2**, we evaluate the correctness of $c_2$ and $c_3$ in Rising-2 triples. Using ground-truth answers and GPT-4o-based annotation (prompt details in Appendix G), we assess whether these steps are logically aligned with the ground-truth answers. As shown in Figure 4(b), $c_2$ in Rising-2 triples is substantially more accurate than $c_3$, confirming that verification in this case often modifies correct reasoning into incorrect steps. These findings support the hypothesis that excessively high $R_{\text{score}}$ values in hallucinated reasoning traces are symptomatic of overthinking—steps that exhibit apparent reasoning depth but in fact reflect spurious or detrimental reasoning.

To address **Q3**, we analyze the relationship between perplexity and $R_{\text{score}}$. Specifically, we randomly sample 200 reasoning steps from ReTruthQA and compute their perplexities as follows:

$$\text{PPL}(c_k) = \exp\left(-\frac{1}{|c_k|} \sum_{t_{m+1}^k \in c_k} \log p\left(t_{m+1}^k \mid t_{\leq m}^k\right)\right), \tag{10}$$

$$\text{PPL}(\mathcal{C}) = \langle \text{PPL}(c_1), \text{PPL}(c_2), \ldots, \text{PPL}(c_K) \rangle. \tag{11}$$

where $p(t_{m+1}^k \mid t_{\leq m}^k)$ denotes the model's predicted probability for token $t_{m+1}^k$ given the prefix $t_{\leq m}^k$ within the reasoning trace.

As shown in Figure 4(c), perplexity and $R_{\text{score}}$ are strongly negatively correlated—steps with higher reasoning depth tend to have lower perplexity, which is intuitive since deep reasoning often yields more predictable outputs. However, when comparing the final step $c_3$ across stable and Rising-2 triples, we find an interesting phenomenon in Figure 4(d): despite having higher $R_{\text{score}}$, $c_3$ in Rising-2 triples has higher perplexity than in stable triples. This suggests that overthinking steps induced by an incorrect verification result in an uncertain or internally unstable generation.

We hypothesize that such overthinking may reflect *spurious verification*—a behavior where the model performs superficial or misguided validation in pursuit of higher reward during RL fine-tuning. This behavior can persist through distillation into smaller models, propagating reasoning hallucinations. Based on this analysis, we identify a third hallucination pattern: **Pattern #3:** Overthinking reasoning steps exhibit a positive correlation between $R_{\text{score}}$ and perplexity (PPL).

**Experimental Validation.** Building on this observation, to further validate whether excessively high reasoning scores reflect overthinking steps, we sampled reasoning steps with scores $\geq 2.5$ (excluding shallow pattern-matching steps) from various data types on R1-7B and used GPT-4o to annotate the correctness of each step (using the same prompt as in Appendix G). We then used the reasoning score to predict step correctness, searching for the optimal F1 threshold in the range 2.5–5 (step size 0.1). Results show that across different datasets, the optimal threshold for F1 is always $\geq 4$, which matches the hyperparameter $\tau$ set in Appendix H. This demonstrates a strong correspondence between excessively high reasoning scores and overthinking steps. Interestingly, this phenomenon aligns with findings in cognitive neuroscience: both insufficient and excessive reasoning can lead to poor decisions (Langley, 1995; Cools & D'Esposito, 2011).

**Table 4:** Optimal threshold $\tau$ for F1 across datasets.

| | MATH | | MultiHopQA | | Science | |
|---|---|---|---|---|---|---|
| | Best $\tau$ | F1 | Best $\tau$ | F1 | Best $\tau$ | F1 |
| Value | 4.0 | 0.7617 | 4.4 | 0.7495 | 4.5 | 0.7821 |

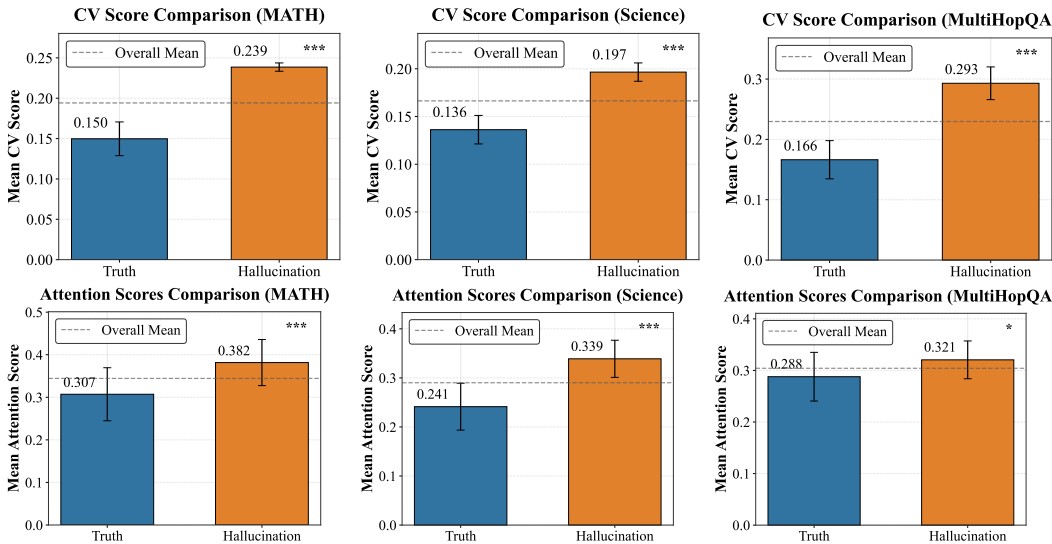

**Figure 6:** Evaluation of Pattern #1 and Pattern #2 on ReTruthQA. Asterisks indicate statistical significance based on a t-test: * for $p$-value $< 0.05$, and *** for $p$-value $< 0.001$.

# G  PROMPT FOR HALLUCINATION PATTERNS ANALYSIS

Prompt for step consistency analysis of **Q1**:

> Please evaluate whether the following reasoning step introduces a new solution approach compared to the preceding steps. Respond with a score of 0 or 1, where:
> 0: The step follows the same solution approach as the previous steps.
> 1: The step explores a new solution approach or direction.
> Reasoning step: {step content}
> Previous steps: {step content}
> Score (0/1):

Prompt for step correctness analysis of **Q2**:

> Please evaluate whether the following reasoning step aligns with the final answer. Respond with a score of 0 or 1, where:
> 0: The step is inconsistent with the final answer.
> 1: The step is consistent with the final answer.
> Reasoning step: {step}
> Final answer: {answer}
> Score (0/1):

# H  MORE RESULTS OF REASONING HALLUCINATION PATTERN ANALYSIS

The hyperparameter settings involved in Section 3.2 are as follows. The constant $r$, which controls the size of the early-step window, is empirically set to $r = 2$. The constant $\eta$, which defines the portion of late reasoning steps, is set to $\eta = 0.75$. The constant $K$, used in computing attention to

earlier steps, is set to $K = 5$. The threshold $\tau$ for identifying potentially overthinking steps is set to $\tau = 4$. These hyperparameters are derived from case analysis and are applied consistently throughout the subsequent reasoning hallucination detection and mitigation experiments.

The validity of Pattern #1 and Pattern #2 is verified across all domains of ReTruthQA, with experimental results shown in Figure 6, where across all three domains, hallucinated reasoning traces consistently exhibit significantly higher CV scores and Attention scores than truthful traces.

## I   EVALUATION AND BASELINE DETAILS OF REASONING HALLUCINATION DETECTION

Based on ReTruthQA, we design two evaluation settings for RHD model: **(1) Binary Detection Setting:** This setting assesses the model's ability to detect hallucinations in individual question-reasoning pairs $(Q, C)$, measuring detection performance using the Area Under the ROC Curve (**AUC**) and Pearson Correlation Coefficient (**PCC**); **(2) Multi-Trace Ranking Setting:** This setting evaluates the model's ability to identify the truthful answer among multiple reasoning traces for the same question $(Q, \{C_1, C_2, \ldots, C_3\})$. We follow the evaluation setup of TruthfulQA-MC (Lin et al., 2021), and report the following metrics: **MC1**: The percentage of instances where the hallucination score of the most hallucinated reasoning trace exceeds that of all truthful traces; **MC2**: The normalized total hallucination score assigned to the hallucinated reasoning traces; **MC3**: The percentage of hallucinated reasoning traces that receive a higher hallucination score than all truthful traces. These metrics collectively measure the ranking quality of hallucination detection in multi-sample generation settings.

For baselines, we consider the following categories: **(1) Ensemble-based self-evaluation methods**, where hallucination scores are obtained through repeated generation, self-verification, or peer voting among LLMs. This category includes ChainPoll (Friel & Sanyal, 2023), LMvLM (Cohen et al., 2023), and SelfCheckGPT (Manakul et al., 2023). **(2) Uncertainty-based methods**, which estimate hallucination likelihood based on model uncertainty, including P(True) (Kadavath et al., 2022), LN-Entropy (Ren et al., 2022), and Perplexity (PPL) (Malinin & Gales, 2020). **(3) Self-awareness-based methods**, which rely on internal model representations to detect hallucinations, such as UQAC (Li et al., 2025b) and EigenScore (Chen et al., 2024). **(4) LLM-as-Critic models**, including `GPT-4o` (Achiam et al., 2023) and `Qwen2.5-32B` (Yang et al., 2024), which act as external evaluators of reasoning traces. **(5) Process reward models**, such as `Qwen2.5-Math-7B-PRM800K` (Zheng et al., 2024a) and `Qwen2.5-Math-PRM-7B` (Zhang et al., 2025), trained with step-level supervision for reasoning evaluation. **(6) Length-based scoring**, motivated by recent findings that longer reasoning traces are more prone to hallucinations (Zeng et al., 2025), we include `Length-Score`, which directly uses the length of the reasoning trace as its hallucination score.

## J   IMPLEMENTATION DETAILS FOR REASONING HALLUCINATION DETECTION

We conduct all experiments on machines equipped with NVIDIA A6000 GPUs and 52-core Intel(R) Xeon(R) Gold 6230R CPUs running at 2.10GHz. We utilize the Huggingface `Transformers` and `TRL` libraries to implement and run our experiments. During response generation, we use random sampling with a temperature of 0.7 and a maximum decoding length of 15,000 tokens for Math tasks and 10,000 tokens for all other tasks. For Reasoning Hallucination Detection (RHD), we perform two-fold validation to select optimal hyperparameters, while baselines are tuned within the ranges specified in their original works. To ensure stability, all randomized experiments are repeated three times and the average results are reported.

We conduct a grid search to identify the optimal reasoning-score weights. Specifically, we search $\alpha_2$, $\alpha_3$, and $\alpha_4$ over the interval $[0, 1]$ with a step size of 0.1, and $\alpha_1$ over $[-1, 1]$ with the same step size. Two-fold cross-validation is used to select the final hyperparameters. For R1-7B, the best weights in the `Math` domain are $\alpha_1 = 0$, $\alpha_2 = 0.4$, $\alpha_3 = 0$, and $\alpha_4 = 0.3$ for the Multi-Trace Ranking setting, and $\alpha_1 = 0$, $\alpha_2 = 0.9$, $\alpha_3 = 0.8$, and $\alpha_4 = 0.4$ for the Binary Detection setting. In the `Science` domain, the best weights are $\alpha_1 = 0.1$, $\alpha_2 = 1.0$, $\alpha_3 = 0$, and $\alpha_4 = 0$ for Multi-Trace Ranking, and $\alpha_1 = -0.4$, $\alpha_2 = 0.9$, $\alpha_3 = 0.5$, and $\alpha_4 = 0.1$ for Binary Detection. In the `MultiHopQA` domain,

**Table 5:** Ablation study of the RHD model on three different domains of ReTruthQA. Each row removes one component of the hallucination score.

| Model | Variant | MATH | | | Science | | | MultiHopQA | | |
|-------|---------|------|------|------|---------|------|------|------------|------|------|
| | | MC1 | MC2 | MC3 | MC1 | MC2 | MC3 | MC1 | MC2 | MC3 |
| R1-7B | RHD | 0.6591 | 0.4765 | 0.5699 | 0.6207 | 0.5448 | 0.6009 | 0.7660 | 0.6255 | 0.7103 |
| | RHD (w/o $\mathrm{Avg}(\mathcal{R}_{\mathrm{score}})$) | 0.6591 | 0.4765 | 0.5699 | 0.6128 | 0.5307 | 0.5934 | 0.7383 | 0.6032 | 0.7082 |
| | RHD (w/o CV Score) | 0.6364 | 0.4663 | 0.5330 | 0.4483 | 0.3862 | 0.4977 | 0.7447 | 0.6043 | 0.6996 |
| | RHD (w/o Attention Score) | 0.6591 | 0.4765 | 0.5699 | 0.6207 | 0.5448 | 0.6009 | 0.6383 | 0.5372 | 0.6123 |
| | RHD (w/o PCC Score) | 0.5909 | 0.3830 | 0.5210 | 0.6207 | 0.5448 | 0.6009 | 0.6809 | 0.5553 | 0.6323 |
| R1-14B | RHD | 0.3692 | 0.3005 | 0.4644 | 0.6667 | 0.4714 | 0.5671 | 0.5785 | 0.4421 | 0.5154 |
| | RHD (w/o $\mathrm{Avg}(\mathcal{R}_{\mathrm{score}})$) | 0.3538 | 0.2867 | 0.4847 | 0.7241 | 0.4609 | 0.5531 | 0.5589 | 0.4284 | 0.5290 |
| | RHD (w/o CV Score) | 0.3692 | 0.2882 | 0.4725 | 0.6470 | 0.4484 | 0.5332 | 0.5455 | 0.4273 | 0.5403 |
| | RHD (w/o Attention Score) | 0.3231 | 0.2692 | 0.4503 | 0.6724 | 0.4511 | 0.5190 | 0.5702 | 0.4322 | 0.5180 |
| | RHD (w/o PCC Score) | 0.3692 | 0.2882 | 0.4725 | 0.6724 | 0.4601 | 0.5683 | 0.5785 | 0.4421 | 0.5154 |

the best weights are $\alpha_1 = 0.4$, $\alpha_2 = 0.1$, $\alpha_3 = 0.6$, and $\alpha_4 = 0.4$ for Multi-Trace Ranking, and $\alpha_1 = 0$, $\alpha_2 = 0$, $\alpha_3 = 0.3$, and $\alpha_4 = 0$ for Binary Detection.

For R1-14B, the best weights in the `Math` domain are $\alpha_1 = 0.3$, $\alpha_2 = 0.7$, $\alpha_3 = 0.1$, and $\alpha_4 = 0.1$ for Multi-Trace Ranking, and $\alpha_1 = 0$, $\alpha_2 = 0.3$, $\alpha_3 = 1.0$, and $\alpha_4 = 0.2$ for Binary Detection. In the `Science` domain, we obtain $\alpha_1 = 0$, $\alpha_2 = 0.5$, $\alpha_3 = 0.5$, and $\alpha_4 = 0.1$ for Multi-Trace Ranking, and $\alpha_1 = -0.2$, $\alpha_2 = 0.2$, $\alpha_3 = 0.9$, and $\alpha_4 = 0.1$ for Binary Detection. In the `MultiHopQA` domain, the optimal weights are $\alpha_1 = 0.7$, $\alpha_2 = 0.9$, $\alpha_3 = 0.1$, and $\alpha_4 = 0.0$ for Multi-Trace Ranking, and $\alpha_1 = 1.0$, $\alpha_2 = 0$, $\alpha_3 = 0.1$, and $\alpha_4 = 0.1$ for Binary Detection.

Candidate reasoning score layers $\mathcal{J}$ are selected from $\{14, 16, 18, 20, 22, 24, 26\}$ for R1-7B and from $\{32, 36, 40, 42, 44, 46\}$ for R1-14B, while attention score layers $\mathcal{L}$ are fixed across models as $\{1, 3, 5, 7, 9, 11, 13\}$. The models used in our experiments, `DeepSeek-R1-Distill-Qwen-7B` and `DeepSeek-R1-Distill-Qwen-14B`, are publicly available at `https://huggingface.co/deepseek-ai/DeepSeek-R1-Distill-Qwen-7B` and `https://huggingface.co/deepseek-ai/DeepSeek-R1-Distill-Qwen-14B`, respectively.

## K  RHD ON LATENT CoT MODELS

To validate whether our proposed hallucination detection framework can be effectively extended to latent Chain-of-Thought (CoT) models, we conducted additional experiments on Huginn-0125, the most mature and open-sourced latent reasoning model currently available (Geiping et al., 2025).

**Application to Huginn-0125.**  The Huginn-0125 model is composed of a prelude block, a core recurrent block (where latent reasoning primarily occurs), and a coda block. We treated each recurrent block as a reasoning step and applied the Reasoning Score to measure reasoning depth, focusing on Pattern #1 (CV score). Due to the model's architectural design, Pattern #2 (AttnScore) and Pattern #3 (PCC) could not be applied. We sampled 100 examples from GSM8K as the test dataset. Since the model itself does not possess self-reflection capability, hallucination labels were assigned based on ground-truth correctness: correct answers were treated as non-hallucinated and incorrect answers as hallucinated. Results in Table 6 show that our method achieved higher detection accuracy of latent reasoning hallucination compared to the perplexity baseline (computed only on output tokens), demonstrating RHD's applicability to latent CoT architectures.

**Comparison with Text-only Detection Methods.**  We further examined whether text-only detection methods can serve as effective alternatives in the latent CoT setting. Specifically, we evaluated Process Reward Model (Qwen2.5-PRM-7B) and ChainPoll. Experimental results in Table 6 indicate that these methods, when applied directly to the final generated text without analyzing the latent reasoning process, underperform compared to our latent reasoning-based approach.

To better understand this gap, we conducted a case study on Huginn-0125 outputs and identified several sources of failure for text-only methods:

- **Format confusion and ambiguous expression.** Consider the following GSM8K problem: Anthony and his friend Leonel read about the importance of keeping pets at home and decided to start adopting cats and dogs from the local rescue center. Anthony has 12 cats and dogs, 2/3 of which are cats. Leonel has half times as many cats as Anthony and seven more dogs than Anthony. How many animals in total do the two have?

  Huginn-0125 produced the following intermediate reasoning:

  > "0.67»10.67 cats. We'll round this to 11 cats for simplicity. Anthony has 12-11=«12-11=1»1 dog. Leonel has half as many cats as Anthony, so he has (1/2)*11=«(1/2)*11=5.5»5.5 cats. We'll round this to 6 cats for simplicity."

  Although the final answer was numerically correct, the text contained artifacts such as "`<<12-11=1>>`" and "`0.67>>10.67 cats`." These inconsistent notations suggest that certain reasoning was carried out in the latent space before being partially surfaced, leading to ambiguous expressions that mislead text-only detectors.

- **Non-linear and highly jumping narration.** Consider the following arithmetic problem: John hits 70% of his free throws. For every foul he gets 2 shots. He gets fouled 5 times a game. How many free throws does he get if he plays in 80% of the 20 games the team plays?

  Huginn-0125 generated:

  > "John gets 5 x 2 = 10 free throws per game. 80% of 20 = 16 games. Then says: 10 fouls per game = 3 x 10 Thus, John gets 16 x 3 = 48 free throws."

  The reasoning begins correctly but suddenly introduces nonsensical statements such as "10 fouls per game = 3 x 10," which are mathematically incoherent. Such non-linear jumps likely originate in the latent CoT process and cannot be effectively diagnosed from surface text alone.

- **Incomplete sentences.** In several cases, Huginn-0125 generated outputs that began with truncated phrases such as "ends each delivered..." without a subject or introductory clause. These malformed sentences indicate leakage of incomplete latent reasoning into surface text, further reducing the reliability of text-only detection models.

These observations highlight that text-only methods are limited in detecting hallucinations when latent reasoning artifacts leak into surface text. By contrast, our approach explicitly analyzes the latent reasoning process, enabling more reliable detection. Importantly, combining the two perspectives proves complementary: empirical results show that integrating our method with ChainPoll achieves the best overall performance.

**Table 6:** Performance of different hallucination detection methods on Huginn-0125. Our RHD approach, when combined with ChainPoll, achieves the best results, indicating complementary benefits.

| Method | AUC | PCC |
|---|---|---|
| LNE | 0.6343 | 0.2312 |
| Qwen2.5-PRM-7B | 0.6460 | 0.2640 |
| ChainPoll | 0.6732 | 0.3074 |
| RHD | 0.6914 | 0.3210 |
| RHD+ChainPoll | **0.7225** | **0.3564** |

## L  ADDITIONAL DETECTION RESULTS ON QWEN3-8B

To examine the generality of our detector beyond the R1-7B/14B backbones, we further evaluate RHD on Qwen3-8B over the RETRUTHQA Math and Science domains. As shown in Figure 7, our method forms the outer envelope across all five metrics (AUC, PCC, MC1/2/3), indicating consistent improvements over diverse baselines. These results mirror the trends reported in the main paper (R1-7B/14B), suggesting that (i) modeling early-stage depth fluctuations, (ii) penalizing misguided backtracking, and (iii) recognizing overthinking (positive RScore–PPL correlation) remain effective

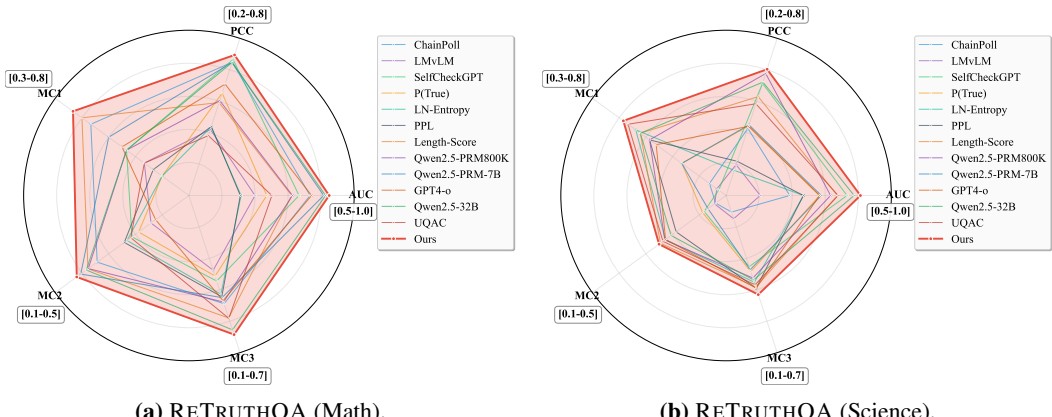

**(a)** RETRUTHQA (Math).      **(b)** RETRUTHQA (Science).

**Figure 7: Performance comparison on `Qwen3-8B`.** Radar plots summarize five metrics: AUC, PCC, and MC1/2/3. Our method (red, dashed outline) consistently dominates the baselines on both domains.

cues on a different backbone and domains. Overall, RHD maintains strong binary detection ability (AUC/PCC) while also excelling in multi-trace ranking (MC1/2/3), reinforcing its robustness across architectures and tasks.

# M  ABLATION STUDY OF RHD

In this section, we analyze the contribution of each module within the RHD model to reasoning hallucination detection. As shown in Table 5, removing any single component leads to a significant performance drop on most datasets in the Reasoning Hallucination Detection task. This validates the effectiveness of adopting a multivariate regression formulation, where all components jointly serve as covariates. Although some coefficients may appear less influential in certain domains, they demonstrate notable impact in others. This observation suggests that different domains exhibit distinct hallucination pattern preferences, further supporting the validity of the empirically discovered patterns, which can be effectively leveraged for reasoning hallucination detection.

Beyond component-level ablations, we also evaluate alternative step-level signals by replacing the Reasoning Score with entropy and variance, resulting in RHD(Entropy) and RHD(Variance). As shown in Table 9, both variants perform substantially worse than the original RHD across MATH, SCIENCE, and MULTIHOPQA. The main reason is that entropy and variance only characterize properties of single distributions, while our approach explicitly models distances between distributions across layers, which is crucial for capturing mechanistic interpretability insights. Furthermore, leveraging the logit lens mitigates the superposition problem in hidden states, enabling more accurate reasoning hallucination detection.

Finally, we analyze the robustness of the shallow-step threshold used in the Attention Score component. In the main method, we adopt the 25% quantile of early reasoning steps as the cutoff for identifying shallow steps, which follows common statistical practice for lower-bound filtering and is consistent with our CV-based fluctuation analysis in Appendix F. To further examine its stability, we vary the threshold over {10%, 20%, 25%, 30%, 40%} and evaluate its impact on MATH–MC3 and MultiHopQA–MC3 under the same setting as Figure 8. Table 8 shows the results.

We observe that performance remains stable across a wide range of threshold values, with the most consistent and balanced results occurring around the 20–30% range. The 25% setting selected in the main paper lies near the empirical optimum and yields strong performance on both benchmarks. These results further confirm that RHD is robust to the choice of threshold and does not rely on fine-grained hyperparameter tuning.

Table 7: Impact of selecting candidate layers from different depth layers of LRMs.

| Layers | Math | | | Science | | | MultiHopQA | | |
|--------|------|------|------|---------|------|------|------------|------|------|
| | MC1 | MC2 | MC3 | MC1 | MC2 | MC3 | MC1 | MC2 | MC3 |
| High | 0.6591 | 0.4765 | 0.5699 | 0.6207 | 0.5448 | 0.6009 | 0.7234 | 0.5957 | 0.6799 |
| Middle | 0.6591 | 0.4765 | 0.5699 | 0.6207 | 0.5448 | 0.6009 | 0.7021 | 0.5862 | 0.6678 |
| Low | 0.6591 | 0.4765 | 0.5699 | 0.6207 | 0.5448 | 0.6009 | 0.7660 | 0.6255 | 0.7103 |

Table 8: Sensitivity of the shallow-step threshold in the Attention Score.

| Threshold | MATH–MC3 | MultiHopQA–MC3 |
|-----------|----------|----------------|
| 10% | 0.5612 | 0.7035 |
| 20% | 0.5683 | 0.7101 |
| 25% | 0.5699 | 0.7103 |
| 30% | 0.5707 | 0.7089 |
| 40% | 0.5521 | 0.6974 |

# N  SENSITIVITY ANALYSIS OF RHD

In this section, we conduct sensitivity analysis experiments to investigate the impact of design choices in RHD. Inspired by the underlying reasoning mechanism, we fix the reasoning score to be extracted from the later layers of LRMs. Our primary focus is on selecting the appropriate layers for computing the attention score. Specifically, we evaluate three different layer groups: shallow layers (1, 3, 5, 7, 9, 11, 13), middle layers (8, 10, 12, 14, 16, 18), and deep layers (14, 16, 18, 20, 22, 24, 26) on R1-7B. The experimental results are shown in Table 7. We observe that, across the `Math` and `Science` domains, the choice of attention layers has limited influence on final performance. In contrast, for the `MultiHopQA` domain, shallow layers yield stronger results, aligning with the mechanistic interpretation that earlier layers are primarily responsible for information transmission. Based on these findings, we select the shallow layers as candidate layers for computing the attention score.

We further perform sensitivity analysis on influential feature weights in RHD across domains. We vary the feature weights in {0.1, 0.3, 0.5, 0.7, 0.9}, and present the results in Figure 8. We observe that most features exhibit an initial increase in performance followed by either a decline or stabilization. The limited variance across settings indicates that the model is not overly sensitive to individual hyperparameter values, demonstrating the robustness and stability of the RHD framework.

For the threshold $\tau$, its selection is based on the analysis described in Appendix F; we performed sensitivity experiments at values [4.0, 4.4, 4.5], and found that the optimal result is achieved at 4.0. Setting the threshold too high improves precision but reduces recall. Encouraging an appropriate depth of reasoning helps the model generalize better, which demonstrates the effectiveness of our chosen hyperparameters.

# O  IMPLEMENTATION DETAILS FOR REASONING HALLUCINATION MITIGATION

We fine-tune the models for reasoning hallucination mitigation using a RL framework with the following hyperparameters: batch size of 8, learning rate of $1.0 \times 10^{-6}$, and 1 training epoch. We enable gradient checkpointing to reduce memory usage. The model is configured with a maximum prompt length of 512 and a maximum completion length of 7680. For parameter-efficient tuning, we adopt LoRA with rank $r = 16$ and $\alpha = 16$, applied to all linear layers (`lora_target_modules=all-linear`). During each training step, we sample 16 generations per query.

The reward function is a weighted sum of three components: (1) an accuracy reward that combines a rule-based parser (Hugging Face, 2025) and LLM-as-a-Judge (Lightman et al., 2023) to determine correctness, addressing the issue where the final answer is correct but fails rule-based extraction

**Table 9:** Ablation results on R1-7B when replacing reasoning score with entropy or variance.

| | Math | | | | | Science | | | | | MultiHopQA | | | | |
|---|---|---|---|---|---|---|---|---|---|---|---|---|---|---|---|
| Method | AUC | PCC | MC1 | MC2 | MC3 | AUC | PCC | MC1 | MC2 | MC3 | AUC | PCC | MC1 | MC2 | MC3 |
| RHD | 0.7978 | 0.4852 | 0.6591 | 0.4765 | 0.5699 | 0.6528 | 0.2662 | 0.6207 | 0.5448 | 0.6009 | 0.7361 | 0.3863 | 0.7660 | 0.6255 | 0.7103 |
| RHD(Entropy) | 0.6523 | 0.2687 | 0.6272 | 0.4293 | 0.5302 | 0.6085 | 0.2289 | 0.5910 | 0.5062 | 0.5836 | 0.6827 | 0.3310 | 0.6170 | 0.5004 | 0.5637 |
| RHD(Variance) | 0.6459 | 0.2657 | 0.6363 | 0.4295 | 0.5257 | 0.5576 | 0.1031 | 0.5172 | 0.4689 | 0.5805 | 0.5866 | 0.1674 | 0.5957 | 0.4776 | 0.5173 |

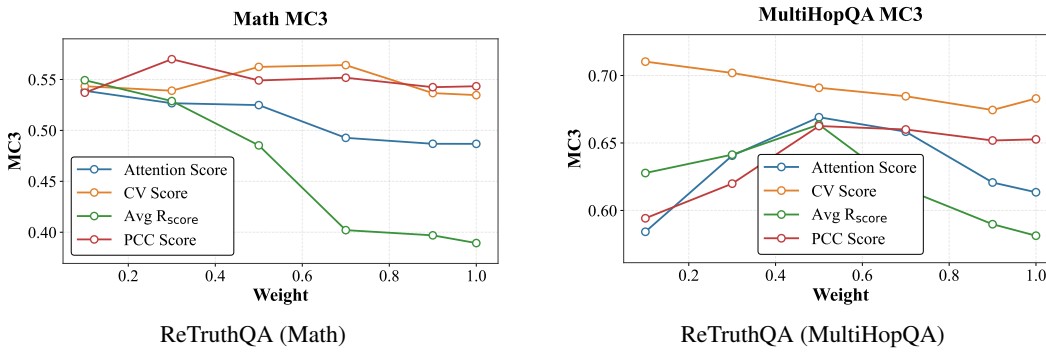

ReTruthQA (Math)             ReTruthQA (MultiHopQA)

**Figure 8:** We conduct a sensitivity analysis of each module in RHD, using R1-7B on the Math and MultiHopQA subsets of ReTruthQA. We vary the weights assigned to different components and observe the resulting performance on the MC3 metric.

(reward = 1 for correct, 0 for incorrect); (2) a **format reward** that ensures adherence to the required reasoning format `<think>\n...\n</think>\n<answer>\n...\n</answer>` (reward = 1 if the format is correct, 0 otherwise); and (3) a **tag count reward** that softly encourages the inclusion of each of the four required tags (`<think>`, `</think>`, `<answer>`, `</answer>`) by assigning 0.25 for each tag present. The reward weights are set to 1.0, 0.1, and 0.1 for the accuracy, format, and tag count rewards, respectively.

For evaluation, we use the same accuracy-based metric as in training, and report results by averaging over four sampled generations per input. The fine-tuned model, `DeepSeek-R1-Distill-Qwen-1.5B`, is publicly available at `https://huggingface.co/deepseek-ai/DeepSeek-R1-Distill-Qwen-1.5B`.

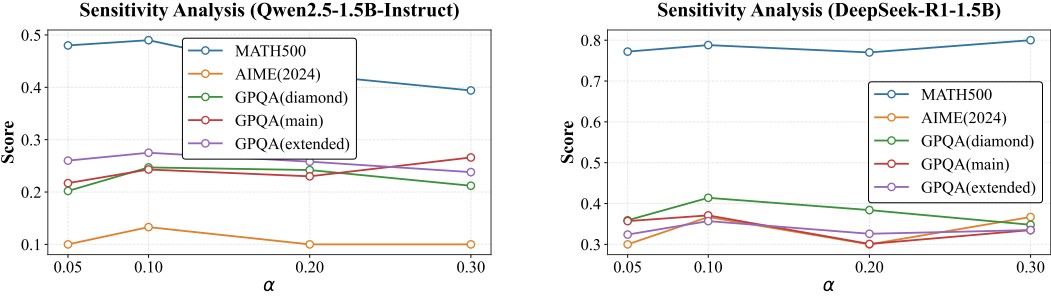

**Figure 9:** We conduct a sensitivity analysis on the weight of the reasoning score reward in GRPO-R, evaluating its impact on the accuracy metric. Experiments are carried out on both Qwen2.5-1.5B-Instruct and DeepSeek-R1-1.5B by varying the weight parameter $\alpha$.

## P   SENSITIVITY ANALYSIS OF REASONING SCORE WEIGHT IN GRPO-R

To investigate the sensitivity of the reasoning score reward weight $\alpha$ in the GRPO-R objective, we conduct experiments on both `DeepSeek-R1-1.5B` and `Qwen2.5-1.5B-Instruct`. We vary $\alpha$ in the range $[0.05, 0.1, 0.2, 0.3]$ and evaluate the models' performance accordingly.

Experimental results in Figure 9 indicate that both models achieve the best average performance when $\alpha = 0.1$. As $\alpha$ increases beyond this value, we observe a gradual decline in performance.

**Table 10:** Sensitivity analysis of threshold $\tau$ across benchmarks.

| Model | MATH500 | AIME(2024) | GPQA(diamond) | GPQA(main) | GPQA(extended) |
|---|---|---|---|---|---|
| GRPO-R $\tau = 4.0$ | 0.788 | 0.367 | 0.414 | 0.371 | 0.357 |
| GRPO-R $\tau = 4.4$ | 0.788 | 0.367 | 0.409 | 0.371 | 0.352 |
| GRPO-R $\tau = 4.5$ | 0.784 | 0.333 | 0.389 | 0.355 | 0.348 |

**Table 11:** Accuracy of distilled models across benchmarks using different sampling strategies. Distillation is performed on Qwen2.5-1.5B-Instruct using reasoning traces from R1-14B.

| Method | MATH500 | AIME (2024) | GPQA (diamond) | GPQA (main) | GPQA (extended) |
|---|---|---|---|---|---|
| Qwen2.5-1.5B-Instruct | 0.466 | 0.100 | 0.202 | 0.197 | 0.211 |
| Random 20% | 0.504 | 0.100 | 0.247 | **0.230** | 0.242 |
| RHD 20% | **0.520** | 0.100 | **0.263** | 0.210 | **0.249** |
| Random 50% | 0.488 | 0.033 | 0.187 | 0.248 | **0.266** |
| RHD 50% | **0.516** | **0.200** | **0.247** | **0.250** | 0.242 |
| 100% | 0.488 | 0.100 | 0.217 | 0.210 | 0.214 |

These results suggest that incorporating the reasoning score reward can effectively mitigate reasoning hallucinations without compromising accuracy, as long as it remains a secondary signal. However, overemphasizing the reasoning score (i.e., assigning it a large weight) can lead to a degradation in the model's ability to optimize for correctness, indicating that the reasoning signal should not dominate the outcome-based reward objective.

## Q  RHD-GUIDED REASONING DISTILLATION

Distilling long-chain-of-thought data from large reasoning models to fine-tune smaller LLMs has become a widely adopted strategy for improving reasoning capabilities (DeepSeek-AI, 2025). However, directly fine-tuning small LLMs on raw LRM-generated data risks transferring undesirable reasoning behaviors such as shallow pattern matching or overthinking, potentially introducing reasoning hallucinations into the smaller models. To address this issue, we propose using the RHD score to rank distillation data and select more truthful samples for training.

The distillation setup uses a learning rate of $5.0 \times 10^{-5}$, batch size of 8, and LoRA applied to all linear layers with parameters `lora_r` = 16 and `lora_alpha` = 16. We use the training data from the hallucination mitigation experiment where `R1-14B` produces correct answers, along with their corresponding reasoning traces and final answers. We then score each reasoning trace using the RHD metric and sort the data in descending order. The top 20% and 50% of ranked samples are distilled into a smaller model, `R1-1.5B`, and compared against randomly sampled subsets of 20%, 50%, and 100% of the same data.

Results, as shown in Table 11, demonstrate that RHD-guided distillation consistently yields better performance across most evaluation benchmarks. In contrast, distillation using 100% of the raw data results in degraded performance, likely due to noise introduced by hallucinated or low-quality samples. These findings validate the effectiveness of RHD in selecting high-quality data and mitigating reasoning hallucinations in downstream small LLMs during the distillation process.

## R  EXTENSION GRPO-R TO OTHER GRPO VARIANTS

Our hallucination mitigation framework in § 4.2 is designed as a general mechanism that can be seamlessly integrated into diverse GRPO variants. By incorporating our mechanistically-inspired step-level reasoning score via potential-based shaping, the framework is orthogonal to existing GRPO improvements and can be applied on top of them without modification.

To further validate this claim, we conducted experiments on **Dr. GRPO** (Liu et al., 2025), a representative variant that modifies the group relative optimization scheme. As shown in Table 12, Dr. GRPO achieves stronger in-domain gains than vanilla GRPO but suffers from reduced robustness on out-of-domain evaluation. Importantly, **Dr. GRPO-R** (our framework applied to Dr. GRPO) consis-

tently improves over Dr. GRPO across both in-domain and out-of-domain settings. These results demonstrate that our framework is compatible with and complementary to existing GRPO variants, highlighting its effectiveness as a general-purpose strategy for mitigating reasoning hallucinations.

**Table 12:** Performance of Dr. GRPO and Dr. GRPO-R across benchmarks.

|            | MATH500 | AIME(2024) | GPQA(diamond) | GPQA(main) | GPQA(extended) |
|------------|---------|------------|---------------|------------|----------------|
| Base       | 0.772   | 0.333      | 0.354         | 0.333      | 0.339          |
| +Dr. GRPO  | 0.778   | 0.367      | 0.364         | 0.333      | 0.342          |
| +Dr. GRPO-R | **0.792** | 0.367    | **0.394**     | **0.364**  | **0.357**      |

In addition to Dr. GRPO, we further validate our hallucination mitigation framework against a widely used step-level reward baseline. Specifically, we adopt a Process Reward Model (PRM) following Shao et al. (2024), where step-level scores are provided by Qwen2.5-PRM-7B. We replace the reasoning score in GRPO-R with PRM scores, denoted as **GRPO+PRM**, for comparison.

Experimental results show that on in-domain MATH datasets, GRPO+PRM achieves results comparable to GRPO-R. However, on out-of-distribution SCIENCE datasets such as GPQA, GRPO+PRM performs noticeably worse than GRPO-R, and in some cases even worse than the base model. This indicates that compared to Qwen2.5-PRM-7B, using the reasoning score as a process reward not only enhances reasoning within the training domain but also generalizes better across domains, underscoring the robustness of GRPO-R. Moreover, incorporating PRM leads to significantly increased training cost. Since PRM itself is a large model, GRPO+PRM requires more training time compared to GRPO-R, further highlighting the efficiency advantage of our approach. Finally, our framework is compatible with PRM. When combining PRM scores with reasoning scores as the final process reward (denoted as **GRPO-R+PRM**), the performance drop on out-of-distribution GPQA benchmarks is effectively alleviated, demonstrating the complementary benefits and generalizability of GRPO-R.

**Table 13:** Comparison between GRPO-R and step-level reward baselines.

| Method       | MATH500  | AIME(2024) | GPQA(diamond) | GPQA(main) | GPQA(extended) |
|--------------|----------|------------|---------------|------------|----------------|
| Base         | 0.772    | 0.333      | 0.354         | 0.333      | 0.339          |
| +GRPO-R      | 0.788    | 0.367      | **0.414**     | **0.371**  | **0.357**      |
| +GRPO+PRM    | 0.780    | 0.367      | 0.343         | 0.330      | 0.333          |
| +GRPO-R+PRM  | **0.792** | **0.400** | 0.409         | 0.373      | 0.355          |

# S    NOTATION SUMMARY

To improve clarity and reproducibility, we provide a comprehensive summary of the key notations used throughout the paper. These notations cover the main components of the Reasoning Score, the RHD detection metric, and the GRPO-R reinforcement learning formulation. The table below consolidates all symbols, their meanings, and where they are introduced in the manuscript.

This notation summary aims to make the paper easier to follow and ensures consistency across the detection and mitigation components of our framework.

# T    COMPLEXITY AND EFFICIENCY ANALYSIS OF RHD

This section presents the theoretical complexity and empirical runtime of the proposed Reasoning Hallucination Detection (RHD) method.

**Theoretical Complexity.**    For a reasoning trace $C = [c_1, \ldots, c_K]$ containing $M$ tokens, the Reasoning Score is computed as the mean Jensen–Shannon Divergence (JSD) between the vocabulary distributions of a small set of later layers ($|J| = 4$–6) and that of the final layer. Each JSD operation

**Table 14:** Summary of the major notations used in the paper.

| Notation | Description |
|---|---|
| $R_{\text{score}}$ | Step-level Reasoning Score measuring reasoning depth via later-layer logit divergence. |
| $CV(C)$ | Coefficient of Variation of early-step reasoning scores, quantifying reasoning fluctuation. |
| $\text{AttnScore}(C)$ | Attention-based metric capturing incorrect backtracking and overthinking behaviors. |
| $\text{PPL}(C)$ | Step-level perplexity sequence used for detecting spurious verification. |
| $\text{PCC}(R_{\text{score}}, \text{PPL}(C))$ | Pearson correlation between reasoning depth and perplexity (Pattern #3 indicator). |
| $\alpha_1, \alpha_2, \alpha_3, \alpha_4$ | Regression coefficients combining hallucination indicators into the final RHD score. |
| $R_{\text{final}}$ | Terminal reward indicating correctness of the final reasoning answer in GRPO-R. |
| $\Phi(s_t)$ | Potential function based on the clipped reasoning score, used for potential-based reward shaping. |
| $\gamma$ | Discount factor controlling reward-shaping dynamics; set to 1 for stability and invariance. |
| $\tau$ | Threshold separating normal vs. overthinking reasoning steps when computing clipped potentials. |
| $r_t, \bar{r}_t$ | Original and shaped step-level rewards in the GRPO-R formulation. |
| $V'(s_t)$ | Value function after reward shaping, ensuring policy-invariance of the optimal solution. |

scales linearly with the vocabulary size $V$, i.e., $O(V)$, consisting of simple element-wise logarithm and multiplication. Therefore, the overall complexity is

$$O(|J| \times M \times V).$$

For comparison, the computational cost of a single Transformer forward pass is

$$O(L \times M \times d^2),$$

where $L$ and $d$ denote the number of layers and hidden dimension. Since the dominant cost comes from quadratic attention and feedforward operations, the linear JSD computation introduces only a negligible constant factor relative to model inference.

Moreover, the design of RHD follows mechanistic interpretability findings that early layers primarily transmit shallow lexical signals. Hence, we compute divergences only over the final 4–6 layers, reducing overall cost by roughly an order of magnitude while preserving its strong correlation with reasoning depth.

**Empirical Runtime.** We further evaluate the practical overhead of RHD by measuring its detection time per query on R1-7B and R1-14B models. All measurements are conducted using HuggingFace Transformers with batch size = 1 on the ReTruthQA (MATH) benchmark.

**Summary.** RHD achieves competitive detection performance while maintaining a sub-second runtime of approximately 0.3 seconds per query. Compared with ensemble-based or reward-model-based detectors, RHD is 10–20× faster due to its lightweight design: it operates directly on cached hidden states and performs JSD computation only on a few later layers. This efficiency makes RHD suitable for large-scale or real-time reasoning hallucination analysis.

## U  FUTURE WORK

Our current framework relies on internal model activations and is thus restricted to open-source LRMs with accessible activations. Extending this line of research to black-box models remains an

**Table 15:** Average detection time per query (seconds) for representative hallucination detection methods.

| Category | Method | R1-7B | R1-14B |
|---|---|---|---|
| Ensemble | ChainPoll | 7.02 | 7.64 |
| Ensemble | LM-v-LM | 12.37 | 11.45 |
| Uncertainty | P(True) | 0.10 | 0.11 |
| Uncertainty | LN-Entropy | 0.05 | 0.09 |
| Self-aware | EigenScore | 0.11 | 0.13 |
| Self-aware | UQAC | 0.45 | 0.76 |
| PRM-based | Qwen2.5-PRM-7B | 10.89 | 10.35 |
| LCM | Qwen2.5-32B | 15.83 | 16.23 |
| Mechanistic | **RHD (ours)** | **0.30** | **0.33** |

important open challenge. Nevertheless, the discovered patterns and metrics may inspire proxy-based extensions that approximate internal reasoning signals without direct access.

Furthermore, our experiments are conducted on moderate-scale models and datasets due to computational constraints. A natural future direction is to scale the proposed framework to larger model families and broader domains, which may provide deeper insights into the universality and robustness of reasoning hallucination mitigation.

