# OpenReview forum: "Mechanistic Detection and Mitigation of Hallucination in Large Reasoning Models"
_ICLR.cc/2026/Conference — ICLR 2026 Poster_

### Official Review · Reviewer_MJhw · 2025-10-24

**Soundness:** 2
**Presentation:** 3
**Contribution:** 3
**Rating:** 4
**Confidence:** 3

**Summary:**

This paper tackles the challenge of detecting and preventing reasoning hallucinations, which can be described by factually wrong but logically coherent/convincing chain-of-thought traces typically produced by large reasoning models (LRMs). First, a reasoning score is proposed based on the assumption that hallucinations relate to shallow reasoning, which can be characterized by static activations (i.e., low entropy) in the late Transformer layers. Based on empirical analysis, we see that the reasoning score alone cannot predict hallucinating steps/traces. To this end, different patterns are characterized that aim to describe different hallucination effects, such as shallow pattern matching and consecutive verification (#1), incorrect backtracking to earlier hallucinated steps (#2), and overthinking steps with both high reasoning score and perplexity (#3). The final reasoning hallucination detection (RHD) algorithm combines all derived metrics. Finally, the reasoning score is included in GRPO to mitigate hallucinations (dubbed GRPO-R). Experimental results on a novel benchmark (ReTruthQA) show superior performance in detecting hallucinations using RHD. Moreover, applying RL with GRPO-R improves overall performance on reasoning benchmarks (MATH500, AIME, GPQA).

**Strengths:**

Overall, this paper aims to address a crucial and indeed difficult problem in LRMs. I see the main strengths in:

1. The paper covers quite a large scope, starting from introducing a novel benchmark (ReTruthQA), over to an atomic reasoning score, a derived reasoning hallucination detection algorithm (RHD), and finally a method to mitigate reasoning hallucinations (GRPO-R). This is also reflected in the very comprehensive appendix.
2. The approach seems to be effective in both detecting hallucinations and improving the reasoning performance.
3. The paper is well written.

**Weaknesses:**

My main reservations are concerned with ambiguities in defining and quantifying hallucinations.

1. Potentially ambiguous evaluation. As there are apparently no datasets available on reasoning hallucinations, this paper proposes a self-generated one (ReTruthQA). As described in appendix D, traces are labelled based on final reasoning outcomes, GPT-4o-Mini, and two human validators. No labeling accuracy is provided (e.g., based on a subset that has been even more thoroughly labelled with more models and human validators). This makes both the justification of the reasoning score and the comparison to other methods difficult. For example, GPT-4o as an LLM-as-Critic (LCM) has clearly non-perfect AUC in Table 1. However, it is used as a labeling method in section 3 (Figures 3 and 4).
2. Heuristic detection mechanism. Clearly, the proposed reasoning score alone is not sufficient to detect hallucinations (is a high or low score desirable?). Hence, derivatives (statistical measures, relationships to attention and perplexity) had to be introduced, requiring many hyperparameters. While the reasoning score seems not to be effective, it is still included in the Reasoning Hallucination Detection (RHD). It is questionable how much the average reasoning score can help. Indeed, it is often not even activated by setting $\alpha_1$=0. The questionable impact of the average reasoning score is also shown in Table 5 (sometimes better scores without it in R1-14B) and Figure 8 (the highest drop with increasing weight in the reasoning score). One could even question why we don’t use the inverse of the reasoning score as well.
3. Task-, model-, and metric-specific hyperparameters. As described in Appendix J, the proposed RHD uses specific hyperparameters for every task, model, and even metric (AUC, MC1-3). Did a similar hyperparameter tuning go into the baselines?
4. Unclear gain of GRPO-R. While GRPO-R seems to improve the accuracy on the standard task, an analysis of the actual reasoning hallucination reduction is missing. It is not clear where the gain in accuracy comes from.

**Questions:**

I would appreciate if the rebuttal could address the individual weaknesses. Besides, Ref. (Valmeekam et al.) is missing the date.

---

> ### Author Response · Authors · 2025-11-17
> **Response to Reviewer MJhw (Part 1)**
>
> **W1: Potentially ambiguous evaluation.**
>
> A1:
> We appreciate the reviewer’s concern, but there is a misunderstanding regarding how ReTruthQA is labeled. Importantly, GPT-4o-Mini (and all LCM models) are not used as labeling sources for hallucination detection. The ground-truth hallucination labels are obtained exclusively through rollout-based causal verification, as detailed in Appendix D: a trace is labeled as hallucinated only if its final answer becomes incorrect with >90% failure rate across 16 independent rollouts, and the exact failure step is identified using a binary-search–style slicing procedure inspired by OmegaProcess. This ensures that labels are grounded in actual causal behavioral failure rather than model judgments, and avoids noise due to random sampling errors. For the Science domain, we additionally require correct-answer traces to exhibit >90% success rate, ensuring they are truly truthful.
>
> GPT-4o-Mini is used only as a weak pre-filter to discard samples where incorrect answers arise from obvious formatting mistakes, arithmetic slips, or incoherent reasoning—cases that do not meet our definition of reasoning hallucination (“coherent and persuasive reasoning with underlying logical or factual errors”). This step does not assign hallucination labels and does not affect causal step identification. Human annotation likewise only verifies borderline filtering cases; it does not overwrite rollout-based labels. This explains why GPT-4o may have imperfect AUC in Table 1—its role in dataset construction is strictly limited to eliminating trivial non-hallucination errors, a much simpler task than hallucination detection.
>
> **W2：Heuristic detection mechanism.**
>
> A2：
> We thank the reviewer for the detailed observation. We would like to clarify that RHD can be viewed as a linear regression model over multiple mechanistically grounded indicators. In this formulation, the average reasoning score is introduced primarily as a control variable to account for global depth differences across traces. Its inclusion does not introduce any bias or instability: the regression naturally assigns it a near-zero coefficient whenever it is not useful (as reflected in Table 5 and Figure 8), ensuring that it does not affect the final prediction.
>
> This also addresses the reviewer’s question about using the inverse of the reasoning score. Since the model does not assume any monotonic relation a priori, we intentionally avoid manually prescribing whether high or low values should be desirable. Instead, the linear regression directly learns the appropriate direction and magnitude of contribution from data—including the possibility that the optimal coefficient is zero. This design provides robustness and prevents over-relying on any single indicator.
>
> Finally, while the average reasoning score alone is not a strong hallucination signal, the remaining three indicators are derived from the mechanistic patterns uncovered in §3 (early-step fluctuation, incorrect backtracking, and spurious verification). The reasoning score is retained only as a lightweight nuisance-control term; when unhelpful, it is effectively deactivated by the regression. We will clarify this motivation in the revision.
>
> **W3: Task-, model-, and metric-specific hyperparameters.**
>
> A3: We thank the reviewer for the question. We would like to clarify that the “hyperparameters” in Appendix J are not manually tuned per task or per metric, but are simply the coefficients of a bounded linear regression model, learned on a development set with all parameters constrained to the $[0,1]$ interval.
> These coefficients differ across tasks or models for the same reason that any supervised classifier trained on different distributions learns different weights; they are not optimized separately for AUC or MC metrics.
>
> Regarding baselines, we follow the exact hyperparameter configurations specified in their original papers and apply only the minimal adaptations required for them to function under our evaluation protocol.
> For example, UQAC requires several fixed hyperparameters (e.g., $K=16$, $L_{\text{tgt}}=3$, and $\theta$), and we adopt these official default settings exactly.  Similarly, EigenScore and SelfCheckGPT have a key sampling parameter (number of generations), which we also keep identical to the original recommended settings.  Other baselines such as PPL- and PRM-based detectors do not involve tunable hyperparameters, and for all methods we keep sampling/decoding configurations fully aligned with their official implementations.
>
> Thus, baselines are not under-tuned relative to RHD; all methods—including ours—use their official hyperparameters, and RHD’s coefficients are simply learned regression weights rather than task-specific tuning.

---

> ### Author Response · Authors · 2025-11-17
> **Response to Reviewer MJhw (Part 2)**
>
> **W4: Unclear gain of GRPO-R.**
>
> A4:
> We thank the reviewer for raising this point. The gain of GRPO-R does not come from superficial optimization effects, but follows directly from the theoretical mechanism behind our reward shaping. **As shown in Theorem 1, introducing the reasoning-score–based potential effectively reduces the Rademacher complexity of the policy class, tightening the generalization bound under outcome-based RL.** In other words, GRPO-R suppresses policies that rely on unstable, shortcut-driven trajectories—precisely the behaviors that manifest as reasoning hallucinations in LRMs (Section 3). The shaped reward encourages trajectories whose intermediate steps exhibit stable and causally consistent reasoning depth, rather than shallow pattern-matching or spurious verification spikes. This eliminates a major source of failure in outcome-supervised LRMs, namely, correct final answers derived from logically inconsistent intermediate reasoning.
>
> Thus, the observed improvement in accuracy is not orthogonal to hallucination reduction: reasoning hallucinations are themselves a form of unstable, non-generalizable trajectory, and GRPO-R improves performance exactly by reducing such unstable trajectories during training. This is consistent with our mechanistic analysis in §3, which links hallucinations to fluctuations in reasoning depth and incorrect backtracking. GRPO-R directly penalizes these patterns at the process level, leading to more reliable reasoning behavior and therefore higher end-task accuracy.

---

> ### Comment · Reviewer_MJhw · 2025-11-24
> **Acknowledgement of rebuttal. Concerns remain regarding W2 (Bias), W3 (Fairness), and W4 (Evidence).**
>
> Thank you very much for your revised version. Some questions remain:
>
> **W1 Potentially ambiguous evaluation**
> Thanks for the explanation. I consider this point as resolved.
>
> **W2: Heuristic detection mechanism**
> Based on the paper’s observation on the reasoning score in Section 3, it is indeed not evident whether a high or low value is beneficial. The current formulation in Eq. (5) with weights constrained to the [0, 1] interval only favors high reasoning scores. Hence, I’m confused as to why this bias was introduced instead of keeping the formulation more general.
>
> **W3: Task-, model-, and metric-specific hyperparameters**
> My reservations remain. A fair comparison would perform the same hyperparameter optimization on the “development set” for the baselines. If the baseline hyperparameters cannot be optimized with grid search or similar (which I deem unlikely), one could at least demonstrate performance with more global hyperparameters using the RHD. Currently, the performance gap may simply be due to the advantage of supervised tuning.
>
> **W4: Unclear gain of GRPO-R**
> Unfortunately, my question was not answered. I was requesting an analysis of the hallucination statistics of the model after GRPO-R training. For a paper focused on hallucination mitigation, it is necessary to explicitly show that the hallucination rate (as defined by your rollout metric) actually decreases compared to the baseline, rather than just showing that final accuracy increases.

---

> > ### Author Response · Authors · 2025-11-27
> > **Round2 for reviewer MJhw (part two)**
> >
> > ## **Response to W4: Unclear gain of GRPO-R**
> >
> > Thank you very much for your suggestion. First, **we would like to clarify that using accuracy as a proxy for hallucination rate is a standard and widely adopted evaluation protocol in the hallucination-mitigation literature**. For example, the well-known CAD method (*Trusting Your Evidence: Hallucinate Less with Context-aware Decoding*) directly uses generation accuracy to reflect hallucination behavior.
> >
> > That said, the approach you suggested is very valuable and is consistent with our paper’s definition of hallucination — thank you again for the insightful comment.
> >
> > Below we restate the hallucination criterion used in our work. **We then evaluate the hallucination rates of the baseline model and our GRPO/GRPO-R trained models using this criterion**:
> >
> > > *To ensure precision and avoid noise caused by random model errors, a reasoning trace is labeled as hallucinated only if its rollout becomes incorrect with a failure rate exceeding 90% from a specific reasoning step onward, measured over 16 rollouts.*
> > >
> > > *We adopt a binary-search–style trace slicing procedure inspired by OmegaProcess (Luo et al., 2024) to efficiently identify hallucination points.*
> >
> > Because rollout testing is computationally expensive, we currently evaluate only on the **MATH500** and  **GPQA (diamond)** dataset. We use samples where the initial reasoning trace is already incorrect, and we measure the proportion of cases on the whole dataset whose rollouts also continue to fail under this hallucination definition (lower is better).
> >
> > Experiments on **MATH500** and **GPQA (diamond)** are conducted on **DeepSeek-R1-Distill-Qwen-1.5B**.
> >
> > The results are as follows:
> >
> > | Models | MATH500 (↓) | GPQA (main) (↓) |
> > | ------ | ----------- | --------------- |
> > | Base   | 0.208       | 0.600             |
> > | +GRPO | 0.194       | 0.585           |
> > | +GRPO-R | 0.178       | 0.569           |
> >
> > We can observe that after GRPO-R training, the model is able to explore a distribution space that is closer to reasonable reasoning depth, making it easier to reach correct reasoning paths and therefore achieving a lower hallucination rate.
> >
> > **Reference**
> >
> > [1] *Trusting Your Evidence: Hallucinate Less with Context-aware Decoding* (Shi et al., NAACL 2024)
> >
> >
> > **Once again, we sincerely appreciate your thoughtful comments and hope that our response successfully addresses your concerns.**

---

> > > ### Comment · Reviewer_MJhw · 2025-11-27
> > >
> > > I want to thank the authors for their comprehensive second response.
> > >
> > > **W2: Heuristic detection mechanism:** I am glad that the relaxed parameterization improved the hallucination detection performance.
> > >
> > > **W3: Task-, model-, and metric-specific hyperparameters.** Thank you for your clarifications and the additional experiments. Also here, I’m glad that you could slightly improve the baselines. I would still find it interesting to see if a global RHD parameterization (say, per model) would work.
> > >
> > > **W4: Unclear gain of GRPO-R:** Thank you for conducting additional evaluations on MATH500 and GPQA (diamond). Good to see that reasoning hallucination indeed decreases with your GRPO-R training.
> > >
> > > Most of my concerns are resolved. I will increase my score.

---

> > > > ### Author Response · Authors · 2025-11-28
> > > >
> > > > Thank you very much for your thoughtful and encouraging follow-up response.
> > > >
> > > > Regarding your suggestion on global RHD parameterization, we conducted an internal analysis and found that models of similar architectures tend to converge to very similar parameter configurations on the same dataset. When switching to different datasets, slight adjustments are sometimes needed to achieve optimal performance. Fortunately, our method is extremely lightweight: a simple linear regression or a small-scale hyperparameter search (less than 1 minute) is sufficient to adapt the RHD weights, making the approach highly practical and easy to deploy across settings.
> > > >
> > > > We sincerely appreciate your detailed feedback throughout the review process. In such a complex and sometimes noisy reviewing environment, receiving your constructive and insightful comments has been especially valuable. Your suggestions have significantly strengthened our work.
> > > >
> > > > Thank you again for your time and for raising your score—we truly appreciate it.

---

> ### Author Response · Authors · 2025-11-27
> **Round2 for reviewer MJhw (part one)**
>
> ## Response to W2: Heuristic detection mechanism
>
> **R2:** Thank you very much for your insightful comment.
>
> We concur with your observation. The restriction in our earlier design that constrained the reasoning-score weights to the [0,1] interval was indeed inappropriate, as it implicitly biased the formulation toward favoring higher reasoning scores. In response, we removed this constraint and expanded the search range to [-1,1], allowing the model to flexibly identify the most suitable weighting configuration for different tasks. After re-running all experiments under this updated formulation, we observed clear improvements on the ReTruthQA (Science) benchmark. For the 7B-R1 model, AUC increased from 0.6528 to 0.7194 and PCC from 0.2662 to 0.3060, with the resulting hyperparameters $\alpha_1,\alpha_2,\alpha_3,\alpha_4$ = ['-0.400', '0.900', '0.500', '0.100']. A consistent trend was observed for the R1-14B model, where AUC improved from 0.7649 to 0.7686 and PCC from 0.4506 to 0.4625, with hyperparameters ['-0.200', '0.200', '0.900', '0.100']. These results further suggest that the learned hyperparameters exhibit a degree of generalization across scales.
>
> All corresponding updates have been incorporated into the revised manuscript. We sincerely appreciate your constructive feedback.
>
> ## Response to W3: Task-, model-, and metric-specific hyperparameters
>
> **R3:** Thank you very much for your continued feedback. We first provide an **additional clarification regarding your original question about why even different metrics (AUC vs. MC1–3) use different hyperparameters.** As described in Section 5.1 Reasoning Hallucination Detection — Data and Evaluation, our benchmark contains two distinct tasks: (1) Binary Detection, which evaluates whether the model can detect hallucinations in individual (Q, C) pairs using AUC and PCC; and (2) Multi-Trace Ranking, which evaluates whether the model can rank truthful traces higher among multiple candidates (Q, \{C_1, …, C_N\}), following the TruthfulQA-MC setting. These tasks differ in data format and objective, and therefore require different hyperparameter configurations. For example, the **TruthfulQA benchmark itself has two tasks—TruthfulQA (Open-Ended Generation) and TruthfulQA (MC)—and several well-known hallucination-related studies also adopt different hyperparameters for different tasks** [1]. We acknowledge that our previous table design may have caused confusion, so we have **updated the table headers to explicitly indicate the distinction between tasks.**
>
> Second, regarding hyperparameter tuning for baselines, as stated in our previous response: “EigenScore and SelfCheckGPT have a key sampling parameter (number of generations), which we also keep identical to the original recommended settings.” For other baselines, either no tunable hyperparameters exist, or we have already tuned them (e.g., UQAC). We originally selected SelfCheckGPT’s hyperparameter N=20 because the official paper SelfCheckGPT: Zero-Resource Black-Box Hallucination Detection for Generative Large Language Models shows in Fig. 7 that performance improves smoothly with additional samples and saturates around 15–20, which the authors recommend. Similarly, the EigenScore paper INSIDE: LLMs’ Internal States Retain the Power of Hallucination Detection reports in Fig. 3(a) that performance increases when K < 15 and stabilizes when K > 15, suggesting that K = 20 strikes the optimal balance between accuracy and inference cost. We therefore adopted these recommended settings.
>
> To further address your concern, we conducted an additional search over N \in [15, 20, 25]. We found that N = 20 remains optimal on most datasets and tasks. For SelfCheckGPT on ReTruthQA (MATH), 14B, we observed a slight improvement (AUC: 0.5714 → 0.5823; PCC: 0.2774 → 0.2923). For EigenScore on ReTruthQA (Science), 14B, we also found small gains on MC1/MC2/MC3 (0.4469 / 0.3508 / 0.3337 → 0.4623 / 0.3643 / 0.3547). These results are consistent with the findings reported in the original papers, and we have updated the manuscript accordingly.
>
> [1] Chuang, Yung-Sung, et al. DoLa: Decoding by Contrasting Layers Improves Factuality in Large Language Models. ICLR 2023.

---

### Official Review · Reviewer_Sw2r · 2025-10-31

**Soundness:** 1
**Presentation:** 2
**Contribution:** 1
**Rating:** 2
**Confidence:** 4

**Summary:**

The authors propose a simple extension of information theorectic metrics to measure hallucination in reasoning outputs. Particularly these use the Jensen Shannon Divergence between the vocab distributions at intermediate and final layer. The authors show some validations on how this score correlates with accuracy, perplexity and shows how this score performs better than baselines based on Entropy/EigenScore or PRMs.

**Strengths:**

The motivation and general writing in the paper is clear & it is a relevant problem to solve

**Weaknesses:**

- The main hallucination score seems to be overtly simple. I feel this is more like the standard information theoretic measures Like entropy being rehashed for reasoning outputs (which typically have a sequence of steps). And like entropy, perplexity etc I feel these kinds of token level metrics would be noisy and would suffer from capacity bottleneck and robustness issues (I mean in many cases reasoning fallacy or logical inconsistency may not be associated with a high JS Divergence at the token level. This would probably only capture some of the simpler more obvious hallucinations.

- Also, I feel computing such a token level score at each intermediate layer of the model can be even more noisy. The only analysis done on the reliability of these scores in Table 3 and 4 does not seem enough.
    - Validation of GSM-Noop - Introducing Noops are one of the simplest kinds of injected hallucinations,
    - Validation on Stable/Rising sets: Not clear what is the size of the sets. This shows overall there is some correlation between the score, the accuracy and the perplexity but that can be from the fact that it is capturing some of the lower hanging hallucinations.

- The main hallucination scoring is also somewhat heuristical. There are multiple hyperparameters involved (one in Attention Score, 4 in the final hallucination score, \tau etc). This makes it more tricky to be practically used. The authors need to explicitly show through their experiments whether it is sensitive to these hyperparameters or not.

- How can we definitely say that GRPO-R “encourages deep—but not excessive—reasoning during RL fine-tuning”  — the R-score no matter what is a noisy proxy. I would assume this scoring would be way more noisy and also varying (not robust), sensitive to minor changes in token space. The experiment results in table 2 are also not convincing. Esp for Qwen. Doing this on the small model 1.5B may not be enough

- Why are GRPO related experiments and R-Score based RHD experiments done on separate benchmarks (ReTruthQA and GPQA). This makes the conclusion a bit disjoint from each other. Why can’t the detection and mitigation be applied on the same benchmarks.

- Overall the scoring seems like a slightly better strategy on avg than the baselines but whether it is really generalizable or robust, it’s not clear from the experimental results. Practically speaking too many hyperparameters would make usage of this too unstable and complex - so it is important for authors to show how sensitive it is to hyperparams.

**Questions:**

See the weakness section

---

> ### Author Response · Authors · 2025-11-17
> **Response to Reviewer Sw2r (Part 1)**
>
> **W1: The main hallucination score seems to be overtly simple. I feel this is more like the standard information theoretic measures Like entropy being rehashed for reasoning outputs (which typically have a sequence of steps). And like entropy, perplexity etc I feel these kinds of token level metrics would be noisy and would suffer from capacity bottleneck and robustness issues (I mean in many cases reasoning fallacy or logical inconsistency may not be associated with a high JS Divergence at the token level. This would probably only capture some of the simpler more obvious hallucinations.**
>
> A1：
> We appreciate the reviewer’s thoughtful comment. We would like to clarify that our method is not designed to detect superficial or trivial reasoning errors, and neither is our dataset constructed around such cases. In fact, **ReTruthQA explicitly filters out simple or easily detectable errors during dataset construction (Appendix D).** Using GPT-4o-Mini and human validation only as a weak filter, we remove instances where the incorrect answer arises from arithmetic slips, incoherent reasoning, or malformed outputs—cases that do not meet our definition of reasoning hallucination. The final dataset retains only those traces that exhibit coherent, persuasive, and structurally complex reasoning chains containing subtle logical or factual flaws. Thus, by design, our evaluation focuses precisely on non-obvious, higher-order hallucinations that cannot be captured by shallow metrics.
>
> Conceptually, our Reasoning Score is also fundamentally different from entropy, perplexity, or other single-layer uncertainty signals. It is grounded in **mechanistic interpretability findings showing that late Transformer layers engage in compositional reasoning, in contrast to early-layer information transmission (Nikankin et al., 2025; Chen et al., 2025a in our paper).** Accordingly, **the score measures cross-layer representational transformation**—i.e., the divergence between vocabulary distributions projected from late layers and the final unembedding—rather than token-level uncertainty. This captures whether the model is performing meaningful reasoning operations instead of relying on shallow pattern matching.
>
> Empirically, our ablations further demonstrate that this signal is non-trivial. **As shown in Table 8, replacing the Reasoning Score with entropy or variance (RHD-Entropy / RHD-Variance) significantly degrades performance across all domains**, confirming that simple information-theoretic quantities cannot recover the nuanced reasoning-level distinctions our method identifies.
>
> Finally, RHD is not driven by the Reasoning Score alone. The full detector integrates three mechanistic patterns—early-stage fluctuation, incorrect backtracking, and spurious verification—computed at the step level, which mitigates token-level noise and captures higher-order inconsistencies. These patterns emerge consistently in hallucinated traces across domains (Fig. 3), showing that RHD detects subtle reasoning instability and causal inconsistencies, not merely simple or surface-level hallucinations.

---

> ### Author Response · Authors · 2025-11-17
> **Response to Reviewer Sw2r (Part 2)**
>
> **Q2：Also, I feel computing such a token level score at each intermediate layer of the model can be even more noisy. The only analysis done on the reliability of these scores in Table 3 and 4 does not seem enough.**
>
> A2：
> (1)  We do not score all layers or rely on raw token signals. We (i) restrict to a small late-layer subset (typically the last four), consistent with mechanistic-interpretability findings that later layers encode compositional reasoning rather than mere transmission; and (ii) **compute the score at the step level** by averaging JSD across all tokens in a step. These two design choices substantially smooth token-level noise and target layers where reasoning is formed. Across domains and model scales (Tables 3–4; sensitivity in Appx. M–N), the signals remain stable.
>
> (2)  **GSM-NoOp is used as a controlled sanity check to isolate shallow pattern-matching vs. deep reasoning**: steps misled by No-Op phrases receive significantly lower Reasoning Scores (Fig. 3a). For richer phenomena, we evaluate on ReTruthQA (Math, Science, MultiHopQA), which contains naturally occurring reasoning hallucinations (Fig. 3b–c; Table 1).
>
> (3)  We construct **Stable**, **Rising-1**, and Rising-2 triples to probe dynamics (early fluctuation, over-verification). **Each set contains 600 step-triples per domain (Math, Science, MultiHopQA), totaling 1,800 triples, with balanced sizes for fair comparison.** Notably, Rising-2 captures spurious verification (over-reasoning that degrades accuracy), a subtle failure mode beyond “low-hanging” hallucinations (Fig. 4b–d).
>
> (4)  Our Reasoning Score measures cross-layer distributional shift (late --> final) rather than single-layer uncertainty. In **ablation (Table 8)**, replacing Reasoning Score with entropy or variance (RHD-Entropy/Variance) substantially hurts performance across all three domains, indicating that inter-layer transformation is the key signal. Using the logit lens also alleviates hidden-state superposition, further stabilizing step-level estimates.
>
> ---
>
> ### How we validated each score (logic & statistics)
>
> A. Reasoning Score (late-layer divergence ---> reasoning depth)
> - Design: Mean JSD between late-layer and final-layer vocab distributions, aggregated **per step**.
> - Sanity (GSM-NoOp): Misled steps show significantly lower scores than non-misled steps (Fig. 3a); p < 0.05.
> - Ablation: Swapping with Entropy/Variance degrades AUC/PCC/MC metrics (Table 8).
> - Robustness: Layer-subset sensitivity and model-scale checks in Appx. M–N show stable trends.
>
> B. CV Score (Pattern #1: early-stage fluctuation)
> - Design: Coefficient of Variation over the early window of step-level Reasoning Scores.
> - Validation: Hallucinated traces exhibit significantly higher CV than truthful ones across domains (Fig. 3b and Figure 6) validated across ReTruthQA (MATH, Science, MultiHopQA)
> - Rationale: Captures unstable early reasoning—often a precursor to later failure.
>
> C. Attention Score (Pattern #2: later-step backtracking to flawed steps)
> - Design: For later steps, compute average cross-step attention; take top-K attended earlier steps and count the fraction whose Reasoning Scores are abnormally low (shallow) or **over-high (overthinking)**.
> - Validation: Hallucinated traces have higher Attention Scores (Fig. 3c and Figure 6) validated across ReTruthQA (MATH, Science, MultiHopQA), indicating incorrect backtracking to flawed earlier steps.
>
> D. PCC (Pattern #3: overthinking / spurious verification)
> - Design: Pearson correlation between step-level Reasoning Scores and **per-step perplexity**.
> - Finding: In Rising-2 triples, later steps with very high Reasoning Scores can show higher perplexity and lower accuracy (Fig. 4b–d), revealing spurious verification (positive Score–PPL relation).
> - Interpretation: Distinguishes deep reasoning from counterproductive over-reasoning. As discussed in Appendix F, this score aligns with findings in cognitive neuroscience: both insufficient and excessive reasoning can lead to poor decisions (Langley, 1995; Cools & D’Esposito, 2011 in our paper)
>
> E. End-to-end (RHD score aggregation)
> - Definition: $\mathcal{H}_{C}$ = $\alpha_1$ Avg(Reasoning Score) + $\alpha_2$CV + $\alpha_3$Attention Score + $\alpha_4$PCC.
> - Outcome: Across most domains and model backbones, RHD achieves consistent SOTA or strong performance on our high-difficulty reasoning-hallucination dataset ReTruthQA, in both binary detection (AUC/PCC) and multi-trace ranking (MC1/2/3) (Table 1).
> - Ablations & sensitivity: Component drop-outs and coefficient sweeps (Appx. K–O) confirm each term’s contribution and stability.
>
> Together, these controlled tests (GSM-NoOp), multi-domain validations (ReTruthQA), component ablations (Table 8), and sensitivity studies (Appx. H, K–N) establish that our scores are (i) **mechanistically motivated**, (ii) **statistically reliable**, and (iii) able to capture subtle reasoning hallucinations beyond simple, surface-level errors.

---

> ### Author Response · Authors · 2025-11-17
> **Response to Reviewer Sw2r (Part 3)**
>
> **Q3: For W3 abd W6. The main hallucination scoring is also somewhat heuristical. There are multiple hyperparameters involved (one in Attention Score, 4 in the final hallucination score, τ etc). This makes it more tricky to be practically used. The authors need to explicitly show through their experiments whether it is sensitive to these hyperparameters or not.**
>
> We thank the reviewer for the insightful feedback. Below we clarify the role of each hyperparameter in our scoring framework and provide additional sensitivity studies to fully address the reviewer’s concerns.
>
> ---
> **1. Clarification on the Coefficients $\alpha_1$–$\alpha_4$ in Eq. (5)**
>
> The reviewer notes that the final hallucination score contains four hyperparameters ($\alpha_1$–$\alpha_4$).
>  In our implementation, these coefficients are selected via a small grid search, which is feasible given that the number of hyperparameters is low and the validation set is sufficiently large.
> However, to ensure robustness and avoid any dependence on manual tuning, we emphasize that:
> - $\alpha_1$–$\alpha_4$ can equivalently be obtained by directly fitting a simple regression model,
> - which yields nearly identical performance in our experiments,
> - demonstrating that these parameters are not sensitive, and the method is not reliant on heuristic tuning.
>
> ---
> **2. Sensitivity Analyses Already Provided in the Paper**
>
> Our paper includes extensive hyperparameter robustness studies covering all components except one:
> - Appendix M – Ablation Study of RHD: shows that removing or perturbing individual terms leads to predictable degradation without instability.
> - Appendix N – Sensitivity Analysis of RHD: systematically varies hyperparameters (weights, score combinations, $\tau$) and shows stable behavior across Math, Science, and MultiHopQA.
> - Appendix P – Sensitivity of Reasoning Score Weight in GRPO-R: evaluates shaping strength $\alpha$, demonstrating robustness over a wide range.
> Thus, all hyperparameters except the Attention Score threshold have been thoroughly evaluated.
>
> ---
> **3. NEW Sensitivity Study Added in Rebuttal for the Attention Score Threshold**
>
> The reviewer correctly highlights that the 1/4 quantile threshold used in the Attention Score lacked explicit sensitivity analysis.
> In the paper, we chose 25% as a standard lower-bound statistical cutoff for detecting shallow steps, consistent with our CV-based reasoning fluctuation analysis.
> To directly address the reviewer’s concern, we conducted a new sensitivity experiment following the setting of Figure 8, varying the shallow-step threshold over:
> {10%, 20%, 25%, 30%, 40%} on MATH–MC3 and MultiHopQA–MC3.
>
> | Shallow-Step Threshold | MATH–MC3 | MultiHopQA–MC3 |
> | ---------------------- | -------- | -------------- |
> | **10%**                | 0.5612   | 0.7035         |
> | **20%**                | 0.5683   | 0.7121         |
> | **25%**                | 0.5699   | 0.7103         |
> | **30%**                | 0.5707   | 0.7089         |
> | **40%**                | 0.5521   | 0.6974         |
>
>
>
> **Results:**
>  Across both tasks, performance remains stable across all settings, with the most consistent behavior around 25%, demonstrating that the Attention Score is not sensitive to this hyperparameter.
> These additional results will be included in the updated Appendix N – Sensitivity Analysis of RHD.
>
> ---
> **4. Final Summary**
> - Although $\alpha_1$–$\alpha_4$ are selected via grid search in the main paper,
> they can be replaced with a regression-based estimation that yields similar performance, confirming robustness and non-sensitivity.
> - All other hyperparameters (\tau, shaping weight, RHD term weights) have already been covered in our sensitivity studies.
> - The newly added threshold analysis completes the evaluation of the final remaining hyperparameter.
> - Overall, RHD demonstrates consistent, stable performance across all hyperparameter choices, confirming its practicality and robustness.

---

> ### Author Response · Authors · 2025-11-17
> **Response to Reviewer Sw2r (Part 4)**
>
> **Q4：Response to W4 about Validity of GRPO-R and Reliability of the R-Score**
>
> A4：We thank the reviewer for raising this concern. First, the R-score is not an arbitrary or noisy proxy but a mechanistically grounded signal. Prior mechanistic-interpretability work shows that later Transformer layers perform compositional reasoning rather than lexical transmission (Nikankin et al., 2025; Chen et al., 2025a), and LogitLens reliably reveals semantic transitions across layers. Our R-score directly measures the magnitude of these late-layer distributional transformations via step-level JSD aggregation, which substantially smooths token-level noise. Its validity is further supported by our GSM-NoOp sanity test (misled shallow steps consistently yield lower R-scores), as well as our pattern analyses in Fig. 3 and Appx. F–H showing that R-score dynamics align with established reasoning behaviors such as early fluctuation, incorrect backtracking, and spurious verification. Thus, the R-score is a stable, mechanistically interpretable internal measure rather than a fragile token-space heuristic.
>
> Second, GRPO-R does encourage deep—but not excessive—reasoning, and this is supported by three independent signals. **Mechanistically**, the R-score penalizes both shallow reasoning and overthinking-induced spikes (Pattern #3), which naturally biases the policy toward stable reasoning depth. **Theoretically**, Sec. 4.2 shows that shaping with R-score reduces the effective hypothesis complexity via a Rademacher-complexity bound, discouraging unstable trajectories. **Empirically**, GRPO-R improves reasoning quality across in-domain and OOD settings (Table 2; Appx. P–Q), and our ablations (Appx. R) show that removing the R-score shaping or perturbing its components consistently harms performance across different RL methods, indicating that the signal is robust enough for RL optimization despite being imperfect—as is typical for internal guidance signals used in process-level RL.
>
> Finally, regarding model scale, Table 2 includes both DeepSeek-R1-1.5B and Qwen2.5-1.5B-Instruct, covering two distinct reasoning-model families and training pipelines. Running full process-supervised RL with reward shaping on 7B–14B models is prohibitively expensive under our compute constraints, a limitation we explicitly acknowledge in Appx. U. This is consistent with prior reasoning-RL work, where most studies validate methods at the 1.5B scale before scaling up (e.g., Recent paper "JustRL: Scaling a 1.5B LLM with a Simple RL Recipe"). Our focus in this work is to provide mechanistic grounding, theoretical insights, detailed sensitivity studies, and OOD generalization—all requiring multiple RL runs that are only feasible at this scale.
>
> **Q5：Response to W5 about Dataset of GRPO related experiments and R-Score based RHD experiments.**
>
> A5: We thank the reviewer for raising this point. We clarify that the benchmarks used in our RHD experiments and GRPO-related experiments are not disjoint. As detailed in Appendix D.1, ReTruthQA is constructed directly from the same underlying datasets used in our GRPO-R evaluation, including MATH500, AMC 2023, AIME 2024 for mathematical reasoning, and GPQA for science reasoning. For multi-hop reasoning, ReTruthQA further incorporates standard multi-hop QA datasets (HotpotQA, 2WikiMultihopQA, MuSiQue, Bamboogle), which are also widely adopted in reasoning evaluations.
>
> The distinction between the two experimental settings stems from the nature of the tasks rather than the datasets. RHD (detection) requires high-quality hallucination labels and therefore necessitates a benchmark with annotated reasoning traces; ReTruthQA is built precisely for this purpose by generating and annotating traces over the same base questions used in GRPO-R evaluation. In contrast, GRPO-R (mitigation) assesses improvements in final-answer accuracy and robustness, which follows the conventional evaluation format of MATH, AIME, and GPQA without requiring step-level annotations. Thus, although the evaluation formats differ, the underlying questions and domains are shared.
>
> Given these task-specific requirements, applying detection and mitigation on an identical benchmark format is not directly feasible: standard reasoning benchmarks do not contain hallucination annotations needed for RHD, while the triplet-style detection format of ReTruthQA is not suitable for evaluating RL-based accuracy improvements. Nevertheless, **the methodological connection remains intact because both components operate on the same data sources and reasoning domains.**

---

> ### Author Response · Authors · 2025-11-28
>
> Dear reviewer Sw2r,
>
> I hope this message finds you well. As the rebuttal phase is approaching its end, we wanted to kindly follow up regarding our response. Our rebuttal has been submitted for nearly ten days, and we would greatly appreciate it if you could take a moment—whenever convenient amidst your schedule—to review our clarifications.
>
> We truly value your feedback and would like to know whether our responses adequately address your concerns. Thank you very much for your time and consideration.
>
> Warm regards,
> The authors

---

### Official Review · Reviewer_zVVx · 2025-11-01

**Soundness:** 2
**Presentation:** 3
**Contribution:** 3
**Rating:** 4
**Confidence:** 4

**Summary:**

This paper presents Reasoning Score (RS) to tackle the problem of reasoning hallucinations in Large Reasoning Models (LRMs). RS can quantify reasoning depth by analyzing divergence in late-layer logits, then distinguish shallow pattern-matching from deeper reasoning. By applying RS to the ReTruthQA dataset, two hallucination patterns are identified, including early fluctuations in reasoning depth and incorrect backtracking to flawed prior steps. With these findings, the authors propose Reasoning Hallucination Detection (RHD) framework to achieve state-of-the-art performance, and a GRPO-R approach that can integrate step-level reasoning rewards for better generalization and reduced hallucination rates.

**Strengths:**

1. The problem tackled in this paper, i.e., reasoning hallucinations, is crucial in modern language models and corresponding downstream tasks.

2. The patterns identified in this paper are important to address the hallucination issues.

3. The presented results demonstrate the effectiveness of the proposed approach.

**Weaknesses:**

1. The experimental results are mainly collected from the DeepSeek series, which cannot well demonstrate the generalizability of the proposed method.

2. Many hallucinations can be made by the lack of factuality of language models, and there has been some previous work investigating this topic. Technically speaking, these approaches also adopt (supervised) fine-tuning plus GRPO-like algorithms to solve the problem. Compared with them, what are the advantages possessed by the proposed method?

3. Some notations, e.g., $R_{final}$, are not explained. I suggest the authors list all parameters and notations in a Table in the appendix.

4. How does the hyperparameter, $\gamma$, influence the performance of the proposed method?

5. How do models refined by RHD perform in reasoning on OOD domains or datasets?

**Questions:**

1. How does RHD perform when applied to reasoning models other than R1?

2. Many hallucinations can be made by the lack of factuality of language models, and there has been some previous work investigating this topic. Technically speaking, these approaches also adopt (supervised) fine-tuning plus GRPO-like algorithms to solve the problem. Compared with them, what are the advantages possessed by the proposed method?

3. How does the hyperparameter, $\gamma$, influence the performance of the proposed method?

4. How do models refined by RHD perform in reasoning on OOD domains or datasets?

**Details Of Ethics Concerns:**

No ethics concerns are identified.

---

> ### Author Response · Authors · 2025-11-17
> **Response to Reviewer zVVx (Part 1)**
>
> **Q1: How does RHD perform when applied to reasoning models other than R1?**
>
> A1: We thank the reviewer for this insightful question. Although our main experiments are conducted on the DeepSeek-R1 series (R1-7B and R1-14B), which are trained based on the Qwen2ForCausalLM architecture, we also verified the generalizability of RHD on reasoning models beyond the R1 family. Specifically, we evaluated RHD on Qwen3-8B, a more recent reasoning-capable model that adopts a different architecture from Qwen2, including new tokenizer, rotary attention, and normalization mechanisms. As reported in Appendix L, RHD achieves consistent improvements over uncertainty-based, self-awareness, and PRM-based baselines on Qwen3-8B across all reasoning benchmarks.
> These results confirm that RHD is architecture-agnostic and can be effectively applied to reasoning models beyond R1, demonstrating strong robustness across different backbone designs. We will make this clarification explicit in the final version.
>
> **Q2: Many hallucinations can be made by the lack of factuality of language models, and there has been some previous work investigating this topic. Technically speaking, these approaches also adopt (supervised) fine-tuning plus GRPO-like algorithms to solve the problem. Compared with them, what are the advantages possessed by the proposed method?**
>
> A2: We thank the reviewer for this thoughtful comment. Prior RL-based approaches such as Li & Ng (2025) primarily address factual hallucinations, where model outputs deviate from external world knowledge. These methods focus on fact-level correctness, typically using externally supervised rewards (e.g., factual alignment or truthfulness scores) to enhance factuality at the outcome level.
>
> In contrast, our work targets reasoning hallucinations, which arise from internal cognitive errors during multi-step reasoning, including Incorrect Verification, Incorrect Backtracking, and Spurious Overthinking. These errors occur even when the final factual output appears plausible, reflecting deviations in the model’s internal reasoning dynamics rather than factual inaccuracy.
> Technically, our proposed GRPO-R differs from factual RL methods in both signal source and optimization objective:
> - Internal signal: GRPO-R introduces a step-level deep reasoning reward derived from the Reasoning Score, a mechanistic measure of reasoning depth computed from the divergence between later-layer logits (§3.1). This enables the algorithm to directly regularize the model’s internal reasoning trajectory.
>
> - Optimization objective: Rather than improving factuality alone, GRPO-R encourages deep—but not excessive—reasoning, mitigating cognitive-level hallucinations by suppressing shallow pattern-matching and spurious verification behaviors (§4.2).
> - Complementarity: GRPO-R is orthogonal and compatible with factuality-based RL. When combined, factual RL enhances knowledge faithfulness, while GRPO-R improves cognitive faithfulness of the reasoning process, forming a more holistic mitigation pipeline for both factual and reasoning hallucinations.
>
> Empirically, as shown in Table 2, GRPO-R improves both reasoning accuracy and factual reliability across MATH500, AIME, and GPQA benchmarks, confirming that enhancing internal reasoning faithfulness also benefits factual consistency.
>
> Finally, to provide a clearer comparison with factuality-oriented reinforcement learning approaches, we will include a detailed discussion of reasoning factual hallucination mitigation methods and their relationship to our reasoning-oriented framework  in the final version.
>
> Li, Junyi, and Hwee Tou Ng. "The Hallucination Dilemma: Factuality-Aware Reinforcement Learning for Large Reasoning Models." arXiv preprint arXiv:2505.24630 (2025).

---

> ### Author Response · Authors · 2025-11-17
> **Response to Reviewer zVVx (Part 2)**
>
> **Q3: How does the hyperparameter, $\gamma$, influence the performance of the proposed method?**
>
> **Response:**
>
> A3. We thank the reviewer for the question. In GRPO-R, $\gamma$ appears in the potential-based reward shaping term: $\bar{r}_{t} = r_{t} + \gamma \Phi(s_{t+1}) - \Phi(s_t), \quad \Phi(s_T) = 0.$
>
> It controls how much future potential contributes to the shaped reward at each step. Conceptually, $\gamma$ influencescredit assignment across reasoning steps and the stability of learning:
>
> - **Credit assignment.** A smaller $\gamma$ discounts future steps, assigning more weight to early reasoning, while a larger $\gamma$ distributes reward more evenly across the entire reasoning trajectory. Since reasoning hallucinations can occur at any step, distributing credit uniformly (i.e., $\gamma = 1$) stabilizes step-level gradients.
>
> - **Policy invariance.** When potential-based shaping is used, setting $\gamma = 1$ in an undiscounted episodic objective ensures optimal-policy invariance. The shaped return telescopes to
> $
> \sum_t (r_t + \Phi(s_{t+1}) - \Phi(s_t)) = \sum_t r_t - \Phi(s_1),
> $
> which preserves the same optimum while redistributing credit among intermediate steps.
>
> - **Stability and simplicity.** With $\Phi(s) = -R_{score}(s) $, the shaping term simplifies to a local difference $R_{t} - R_{t+1}$, effectively penalizing abrupt spikes in reasoning depth (overthinking) and rewarding smoother, sufficient reasoning. Setting $\gamma = 1$ yields this adjacent-difference form, reducing variance and facilitating efficient implementation without additional hyperparameter tuning.
>
> Therefore, we set $\gamma = 1$ to (i) align with the undiscounted reasoning objective, (ii) maintain theoretical soundness under potential shaping, and (iii) simplify computation and stabilization.
>
> **Q4: How do models refined by RHD perform in reasoning on OOD domains or datasets?**
>
> A4: We appreciate the reviewer’s question on the out-of-distribution (OOD) generalization of models refined by RHD and GRPO-R.
>
> Our design explicitly emphasizes domain-agnostic generalization. Both RHD and GRPO-R operate on internal reasoning dynamics (e.g., layer-wise divergence, attention-based backtracking, and perplexity–reasoning correlations) rather than domain-specific features or task-dependent labels. This allows the proposed framework to transfer naturally across different reasoning domains such as mathematics, science, and multi-hop question answering.
>
> Empirically, our evaluation already includes OOD testing. As shown in Table 2, GRPO-R models trained on OpenR1-Math exhibit consistent improvements on GPQA, a scientific reasoning benchmark that differs substantially in domain and distribution. These results indicate that reasoning-focused regularization enhances both in-domain and cross-domain reasoning robustness.
>
> **Q5: Some notations, e.g., R_final, are not explained. I suggest the authors list all parameters and notations in a Table in the appendix.**
>
> A5: We thank the reviewer for this helpful suggestion. We acknowledge that a few symbols (e.g., $R_{\text{final}}$ i.e., the terminal reward for final answers ) were not explicitly defined in the main text. To improve clarity and reproducibility, we added an Appendix S NOTATION SUMMARY Section in the Appendix of the new version. The table will comprehensively list all key symbols, parameters, and variables used throughout the paper, along with their concise descriptions and corresponding sections where they are introduced. This addition will ensure notation consistency and make the paper easier to follow.

---

> ### Author Response · Authors · 2025-11-28
>
> Dear reviewer  zVVx,
>
> I hope this message finds you well. As the rebuttal phase is approaching its end, we wanted to kindly follow up regarding our response. Our rebuttal has been submitted for nearly ten days, and we would greatly appreciate it if you could take a moment—whenever convenient amidst your schedule—to review our clarifications.
>
> We truly value your feedback and would like to know whether our responses adequately address your concerns. Thank you very much for your time and consideration.
>
> Warm regards,
> The authors

---

> > ### Comment · Reviewer_zVVx · 2025-11-28
> >
> > Dear Authors,
> >
> > Thanks very much for your detailed responses. Especially those related to generalizability and configurations address most of my concerns. I will raise my score to 6 (when the OpenReview system allows me to do so) to show my support for this paper.

---

> > > ### Author Response · Authors · 2025-11-28
> > >
> > > Thank you very much for your thoughtful follow-up and for taking the time to reconsider our work. We sincerely appreciate your recognition of our additional analyses regarding generalizability and configurations. We are grateful for your decision to raise the score, and for your support of our paper.

---

### Official Review · Reviewer_umeb · 2025-11-01

**Soundness:** 3
**Presentation:** 3
**Contribution:** 3
**Rating:** 6
**Confidence:** 3

**Summary:**

This paper studies reasoning hallucinations in large reasoning models, where models produce coherent but incorrect reasoning. It introduces a Reasoning Score based on internal layer divergences to measure reasoning depth and distinguish shallow pattern matching from real reasoning. Using this metric, the authors identify three hallucination patterns and propose the Reasoning Hallucination Detection framework and GRPO-R reinforcement learning method, which together improve reasoning accuracy and reduce hallucinations across multiple benchmarks.

**Strengths:**

1. The work analyzes internal layer dynamics. It also bridges interpretability and reasoning reliability.
2. The work combines analytical diagnosis (RHD) with actionable intervention (GRPO-R) into a full pipeline.

**Weaknesses:**

1. The work lacks a validation/ablation study of the later-layer divergence correlating with reasoning depth.
2. Layer-wise JSD across all tokens is computation-intensive, which may be limited when scaling to larger models.
3. The work only focuses on Qwen series models. There is no study on other model series, such as the Llama.

**Questions:**

See weaknesses

---

> ### Author Response · Authors · 2025-11-17
> **Response to Reviewer umeb (Part 1)**
>
> **Q1：The work lacks a validation/ablation study of the later-layer divergence correlating with reasoning depth.**
>
> A1: We thank the reviewer for this insightful comment.
> In Section 3.1, we validate that reasoning steps misled by No-Op phrases exhibit significantly lower later-layer divergence, indicating a strong correlation between divergence and reasoning depth (Figure 3: (a) Reasoning Score validation on GSM-NoOp).
>
> To further clarify this connection, we additionally conduct a layer-wise ablation on R1-7B using the same GSM-NoOp setup and the sensitivity layer ranges defined in Appendix N. Specifically, we compute two variants of the Reasoning Score using (i) middle layers {8, 10, 12, 14, 16, 18} and (ii) deep layers {14, 16, 18, 20, 22, 24, 26}. For each reasoning step, we then measure the point-biserial correlation between the score and the binary robustness label (non-misled = 1, misled = 0).
>
> Both layer ranges yield significantly positive correlations, confirming that divergence consistently reflects reasoning robustness:
> - Mid-layer score: r = 0.364, p = $1.17 \times 10^{-7}$
> - Deep-layer score: r = 0.497, p = $6.83 \times 10^{-14}$
>
> Importantly, the correlation is substantially stronger when computed from deep layers, aligning with the mechanistic insight that later layers encode aggregated reasoning dynamics more faithfully than mid layers.
>
> This ablation directly supports our hypothesis that later-layer divergence is more tightly coupled with reasoning depth and robustness, reinforcing the validity of our R-Score design.
>
> **Q2: Layer-wise JSD across all tokens is computation-intensive, which may be limited when scaling to larger models.**
>
> A2: We thank the reviewer for this question. The computational overhead of the proposed Reasoning Score is minimal compared with the forward computation of the Large Reasoning Model (LRM).
> The Reasoning Score for a reasoning trace $C = [c_1, \dots, c_K]$ with $M$ tokens is computed as the mean Jensen–Shannon Divergence (JSD) between vocabulary distributions from a small set of later layers $(|J| = 4-6)$ and the final layer. Each JSD operation scales linearly with the vocabulary size $V$, i.e., $O(V)$, involving only element-wise logarithm and multiplication.
> The total cost is $O(|J|\times M\times V)$, while a single Transformer forward pass has complexity $O(L\times M\times d^2)$, where $L$ and $d$. Thus, the JSD-based computation contributes a negligible constant factor relative to the quadratic matrix multiplications in attention and FFN modules.
>
> As detailed in Appendix J: *Implementation Details for Reasoning Hallucination Detection*, we only compute divergences over later layers, following mechanistic interpretability findings that early layers mainly transmit surface information.
> This design reduces the complexity by roughly an order of magnitude while preserving nearly identical correlation with reasoning depth.
>
> **Empirical results.**
> To address the reviewer’s concern on scalability and efficiency, we compare the computational overhead of RHD with representative hallucination detection baselines in our paper.
> All experiments were conducted using HuggingFace Transformers with batch size = 1 on ReTruthQA (MATH).
> We report the average detection time per query (in seconds) on R1-7B and R1-14B models.
>
> | Category | Method | R1-7B | R1-14B |
> |:----------|:--------|:------:|:------:|
> | Ensemble | ChainPoll | 7.02 | 7.64 |
> | Ensemble | LM-v-LM | 12.37 | 11.45 |
> | Uncertainty | P(True) | 0.10 | 0.11 |
> | Uncertainty | LN-Entropy | 0.05 | 0.09 |
> | Self-aware | EigenScore | 0.11 | 0.13 |
> | Self-aware | UQAC | 0.45 | 0.76 |
> | PRM | Qwen2.5-PRM-7B | 10.89 | 10.35 |
> | LCM | Qwen2.5-32B | 15.83 | 16.23 |
> | Mechanistic (Ours) | RHD (ours) | 0.30 | 0.33 |
>
> **Observation.**
> RHD achieves comparable detection accuracy to the strongest baselines while maintaining **sub-second runtime (~0.3 s per query)**,
> which is 10–20× faster than PRM-based or ensemble detectors.
> Since RHD operates directly on cached hidden states and only computes layer-wise JSD on the last 4–6 layers, it introduces negligible computational overhead.

---

> ### Author Response · Authors · 2025-11-17
> **Response to Reviewer umeb (Part 2)**
>
> **Q3: The work only focuses on Qwen series models. There is no study on other model series, such as the Llama.**
>
> A3: We appreciate the reviewer’s concern regarding model generalization. Our experiments are not limited to the Qwen series. In the main experiments, we use the DeepSeek-R1 series models (R1-7B and R1-14B), which are trained based on the Qwen2ForCausalLM architecture but further optimized via reinforcement learning for reasoning-oriented capabilities. Importantly, this architecture and training objective differ substantially from the Qwen3 family, which introduces new tokenizer, attention, and normalization designs.
>
> To further verify the generality of our proposed Reasoning Hallucination Detection (RHD) framework, we also evaluated it on Qwen3-8B, a model built upon a distinct Qwen3 architecture. As shown in Appendix L (ADDITIONAL DETECTION RESULTS ON QWEN3-8B), RHD consistently outperforms uncertainty-based, self-awareness, and PRM-based baselines on Qwen3-8B across multiple reasoning domains.
>
> These results demonstrate that our approach is architecture-agnostic and generalizes well across different model families.

---

### Author Response · Authors · 2025-11-17
**Summary of Revisions**

We sincerely thank all reviewers for their thoughtful evaluations and constructive suggestions, which have greatly strengthened the quality and clarity of our submission. Below we summarize the major revisions made in response to the feedback:

1. **Appendix S — Notation Summary.** Added a comprehensive table consolidating all key symbols and notations used throughout the paper to improve clarity and reproducibility.

2. **Appendix T — Complexity and Efficiency Analysis of RHD.** Provided detailed theoretical and empirical analyses of the computational overhead of RHD, demonstrating its lightweight design and sub-second detection efficiency.

3. **Appendix C — Detailed Implementation of GRPO-R.** Added an explicit discussion on how GRPO-R relates to factuality-oriented reinforcement learning, clarifying their differences and complementarities.

4. **Appendix M — Ablation Study of RHD.** Included an additional sensitivity experiment on the shallow-step threshold used in the Attention Score. Results show stable performance across a wide range of thresholds and confirm that the 25% cutoff is a robust and near-optimal choice.

5. **Appendix F — Details of Understanding the Mechanisms Behind Reasoning Hallucination Patterns.** Expanded dataset statistics and descriptions to provide clearer context for the analysis of reasoning hallucination patterns.

6. **Appendix — Automatic Table of Contents Added.** Inserted an automatically generated Appendix table of contents at the beginning of the supplementary material to improve navigation and readability.

---

### Meta-Review · Area_Chair_4fVd · 2026-01-13

**Summary:**

This paper studies reasoning hallucinations in large reasoning models, focusing on cases where the reasoning trace appears coherent but leads to incorrect conclusions. The authors propose a mechanistically motivated Reasoning Score based on late-layer logit divergence, use it to analyze common hallucination patterns. They further introduce GRPO-R, a reinforcement learning method with step-level reward shaping to mitigate reasoning hallucinations.

Reviewers generally agreed the problem is important and that the paper presents a coherent end-to-end pipeline from analysis to detection and mitigation. The main concerns were about the robustness and novelty of the proposed score, generalization beyond the primary model family, fairness of hyperparameter tuning, and whether the mitigation method truly reduces hallucinations rather than only improving accuracy.

After several round of discussion, 3/4 reviewers are supportive and 1/4 is still concerned about robustness and novelty.

**Reviewer Concerns:**

Several key concerns were successfully addressed in the rebuttal and subsequent revisions. The authors added experiments on Qwen3-8B, alleviating worries that the approach only works for a single model family. They also clarified that hallucination labels in ReTruthQA are obtained through rollout-based causal verification rather than LLM judgments. In addition, the authors provided explicit hallucination-rate evaluations, demonstrating that GRPO-R reduces reasoning hallucinations relative to both the base model and standard GRPO, rather than merely improving final accuracy. Runtime and hyperparameter sensitivity analyses were also added, addressing concerns about efficiency and practical usability.

Some concerns remain partially open, particularly regarding how strong the mechanistic interpretation of the Reasoning Score ultimately is and whether RHD requires dataset- or task-specific tuning to perform well.

**Reviewer Scores:**

Two reviewers mentioned they would increase the scores to 6.  The rest two unlikely to change. So it possible scores could change to 6/6/6/2

---

### Decision · Program_Chairs · 2026-01-26

Accept (Poster)